

# Hydrodynamics without boost-invariance from kinetic theory: From perfect fluids to active flocks

**Kevin T. Grosvenor[1], Niels A. Obers[2,3] and Subodh P. Patil[4]**

**1** National Institute of Physics, University of the Philippines,
Diliman, Quezon City 1101, Philippines
**2** Niels Bohr International Academy, The Niels Bohr Institute, University of Copenhagen,
Blegdamsvej 17, DK-2100 Copenhagen Ø, Denmark
**3** Nordita, KTH Royal Institute of Technology and Stockholm University,
Hannes Alfvéns väg 12, SE-106 91 Stockholm, Sweden
**4** Instituut-Lorentz for Theoretical Physics, Leiden University,
2333 CA, Leiden, The Netherlands

## Abstract

We derive the hydrodynamic equations of perfect fluids without boost invariance [1] from kinetic theory. Our approach is to follow the standard derivation of the Vlasov hierarchy based on an a-priori unknown collision functional satisfying certain axiomatic properties consistent with the absence of boost invariance. The kinetic theory treatment allows us to identify various transport coefficients in the hydrodynamic regime. We identify a drift term that effects a relaxation to an equilibrium where detailed balance with the environment with respect to momentum transfer is obtained. We then show how the derivative expansion of the hydrodynamics of flocks can be recovered from boost non-invariant kinetic theory and hydrodynamics. We identify how various coefficients of the former relate to a parameterization of the so-called equation of kinetic state that yields relations between different coefficients, arriving at a symmetry-based understanding as to why certain coefficients in hydrodynamic descriptions of active flocks are naturally of order one, and others, naturally small. When inter-particle forces are expressed in terms of a kinetic theory influence kernel, a coarse-graining scale and resulting derivative expansion emerge in the hydrodynamic limit, allowing us to derive diffusion terms as infrared-relevant operators distilling different parameterizations of microscopic interactions. We conclude by highlighting possible applications.

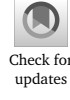

# 1  Introductory remarks

Field theories with Lorentz-invariant interactions can be shown to reproduce the Boltzmann equation and Lorentz boost-invariant hydrodynamics at large occupation [2], where the familiar Navier-Stokes equations result in the non-relativistic limit. However, many interesting systems in nature lack boost invariance, be it Lorentzian or Galilean, due to the presence of a preferred frame. A variety of theoretical and phenomenological motivations have led to the development of field theories where boost symmetries are kinematically absent (as opposed to spontaneously broken[1]) in order to explore the possibility that a preferred frame may be a defining property of the system in question [8–12], which immediately leads to the question of what the corresponding hydrodynamic limit should be.

As elaborated upon in [1], first-principles reasoning leads one directly to a formulation of boost non-invariant hydrodynamics that features important qualitative differences relative to boost-invariant cases. Foremost, conservation of an appropriately defined stress-energy tensor implies that the Euler equation for perfect fluids gets modified according to

$$\partial_t \mathbf{v} + (\mathbf{v} \cdot \nabla)\mathbf{v} = -\frac{\nabla P}{\rho} - \frac{\mathbf{v}}{\rho}\big[\partial_t \rho + \nabla \cdot (\rho \mathbf{v})\big], \tag{1}$$

where the quantity $\rho$ is the so-called *kinetic mass density* that in general depends on $v^2$ (assuming rotational invariance is preserved), and where the equilibrium distribution is obtained from the generalized grand canonical partition function $\mathcal{Z}$ and corresponding density matrix $\rho_\beta$:

$$\mathcal{Z} = \mathrm{Tr}\left(e^{-\beta(\hat{H} - \mu\hat{N} - \hat{\mathbf{v}}\cdot\hat{\mathbf{P}})}\right), \qquad \rho_\beta \equiv \frac{e^{-\beta(\hat{H} - \mu\hat{N} - \hat{\mathbf{v}}\cdot\hat{\mathbf{P}})}}{\mathcal{Z}}. \tag{2}$$

---

[1]Whereby the state characterizing the system breaks boost invariance even if the kinematics of the underlying theory retains the invariance – cf. [3,4], which studies how this can be effected via non-linearly realizing spacetime symmetries and symmetry breaking on fields. The latter closely tracks the treatment of the spontaneous breaking of internal symmetries [5,6] (with notable differences nicely summarized in [7]).

The key physical input that leads to the above is the fact that the absence of boost invariance implies that momentum can in principle be exchanged with the reservoir, and therefore necessitates an additional thermodynamic potential which we label $\hat{\mathbf{v}}$. Further details can be found in [1], which we recap in Appendix A where a symmetry based argument for the necessity of introducing a velocity potential in the absence of boost invariance is presented (see also [13–16] for the study of leading-order transport phenomena, and [17] for a detailed study of finite temperature effects and applications to active matter systems).

The fact that one is forced to factor in momentum exchange with the reservoir if one is to arrive at a first-principles understanding of a generic boost non-invariant system – a flock of birds being the most relatable example – should be immediately apparent to anyone who has ever seen a flock change direction mid-air. By flapping their wings, momentum is transferred to the reservoir (air in this case), allowing birds to accelerate and change their direction of motion. Although obvious to the point that would justify it being taken as a given, such a starting vantage has yet to be formalized and developed in the context of flocking theory, particularly concerning active flocks where the individual birds (or boids) can themselves act as energy, momentum, and entropy sources and sinks, which can be implemented via collision terms consistent with these assumptions in any kinetic theory derivation. For the unfamiliar: the term boid originates from the name of an artificial life program [27], where it's a portmanteau of bird-oid for bird-like objects. It has also been noted to be the New Yorker pronunciation for 'bird' [28].

The possibility of exchanging momentum with the reservoir also forces us to deal with the fact that inertial mass, as defined as resistance to acceleration under the application of external forces, will itself depend on the state of motion of the body in question no matter its velocity. Therefore it is natural to expect that the relation between momentum and velocity will be modified in a manner such that a *kinetic mass density* $\hat{\rho}$ defines the relation

$$n\hat{\mathbf{p}} = \hat{\rho}\hat{\mathbf{v}}, \tag{3}$$

where $n$ is the particle number density and $\hat{\rho}$ in general is $v^2$-dependent.[2] The velocity dependence of the kinetic mass density encodes the nature of the boost non-invariant kinematics of the underlying interactions.

In what follows, we show how any parameterization of the velocity dependence of the kinetic mass density in the kinetic theory treatment forces relations between coefficients of any derivative expansion in the hydrodynamic limit, which under various assumptions reduces to

$$\partial_t \mathbf{v} + (\mathbf{v} \cdot \nabla)\mathbf{v} = -\frac{\nabla P}{\rho} - \frac{\mathbf{v}}{\rho}\left[\partial_t \rho + \nabla \cdot (\rho \mathbf{v})\right] - \frac{\mathbf{v} - \mathbf{v}_{eq}}{\tau}, \tag{4}$$

where implicit in the above is a presumed Bhatnagar-Gross-Krook-Welander (BGKW) form for the collision functional [18, 19], and where inter-boid forces are presumed to vanish. Specifically, in Section 4 we introduce the notion of an *equation of kinetic state* which parameterizes the non-conservation of the kinetic mass density[3] due to momentum transfer with the reservoir, whose physical interpretation is further elaborated upon in Appendix E. It is the parameterized departure of the equation of kinetic state from the boost-invariant limit that forces relations between various hydrodynamic coefficients.

We then go on to elaborate on how the hydrodynamics of flocks [20–22] (see also [23–26] for comprehensive reviews) can be reproduced from (4). In relating different coefficients of the derivative expansion of the former, we arrive at a symmetry-based understanding as to why certain coefficients of Toner-Tu theory [20–22] will naturally be close to unity, and why

---

[2]In what follows, hats are placed on quantities which are generically functions of momentum and position. Once expectation values over momenta are taken, the same quantities will be denoted without hats.

[3]Non conservation of the kinetic mass density results in non-vanishing terms from the square brackets in (4).

others have a propensity to be small, if not vanish outright. We thus arrive at a complementary understanding of the various salient features expected of flocks from a minimal set of physical ingredients. In particular, the drift term encodes the expected relaxation to equilibrium of any flock to co-move with the environment, where equilibrium is further stipulated by detailed balance with respect to momentum transfer.

By repeating the derivation that led to (4) in the presence of inter-boid forces, which we recast in terms of an influence kernel (cf. Appendices B and C), one finds additional contributions to the modified Euler equation:

$$\partial_t \mathbf{v} + \left(\mathbf{v}\cdot\nabla\right)\mathbf{v} = -\frac{\nabla P}{\rho} - \frac{\mathbf{v}}{\rho}\left[\partial_t\rho + \nabla\cdot(\rho\mathbf{v})\right] - \frac{\mathbf{v}-\mathbf{v}_{\text{eq}}^{\text{F}}}{\tau^{\text{F}}} + D_T\nabla^2\mathbf{v} + D_B\nabla\left(\nabla\cdot\mathbf{v}\right), \qquad (5)$$

where the superscripts "F" on the relaxation time and the environment equilibrium velocity are to distinguish them from the corresponding quantities in the force-free expression. The additional diffusion terms – whose coefficients are labeled to coincide with the notation of [22] – arise in the hydrodynamic limit from a derivative expansion for the influence kernel, which we understand in terms of long-wavelength relevant operators that parameterize the details of the inter-boid forces at the microscopic level. In certain limits, these can be viewed as parameterizing different viscosities in the hydrodynamic limit, but are to be viewed more generally in our discussion. In principle, the kinetic theory treatment allows one to compute all transport phenomena, and we conclude our treatment by highlighting possible applications and open questions.

To outline the discussion to follow: Section 2 rederives the perfect fluids formalism of [1] from a kinetic theory treatment. For clarity of discussion, we do this first by following the standard derivation of the Vlasov hierarchy in the absence of inter-particle/boid forces, but with a collision functional relevant to boost non-invariant systems. In 3 we incorporate forces, and discuss how a derivative expansion naturally arises in the hydrodynamic limit with the introduction of an influence kernel. We pause here to review how the derivative expansion by its very nature encodes long-wavelength averages of stochastic fluctuations, obviating the need to explicitly introduce noise terms to model random forces at the kinetic theory level. In Section 4 we reproduce the Toner-Tu hydrodynamic theory of flocks, uncover relations between its various coefficients, and elaborate on why departures from the boost-invariant limit dictate that coefficients of certain operators tend to be close to unity, and others close to vanishing, and understand the diffusion coefficients in terms of long-wavelength parameter relevance. Along the way, we pause to discuss the physical meaning and implications of our findings with various details and elaborations deferred to the appendices, and conclude by discussing possible applications and future directions that warrant further study.

In what follows, we refer to the constituents of the kinetic theory interchangeably as either particles or boids, where the latter is conventional terminology from flocking theory. Although the latter application is what we primarily focus on in this paper, we stress that the formalism developed in what follows is applicable in principle to any boost non-invariant fluid, see e.g. [29–35], which include a diverse range of phenomena across multiple classes of physical systems.[4]

---

[4]Many of these works include critical systems that have an extra (Lifshitz) scaling symmetry, characterized by a dynamical exponent $z$. Indeed, [1] has proven a no-go theorem stating that if such systems allow for a fluid description, they cannot exhibit boost symmetry when $z \neq 1, 2$. A field-theoretic version of this no-go theorem was presented in Ref. [36].

## 2 From kinetic theory to boost non-invariant hydrodynamics

We begin our treatment by assuming the existence of a distribution function $f(t, \mathbf{x}, \hat{\mathbf{p}})$, which is a function of time, space, and momentum. More precisely, $f$ counts the number of single-particle states. We assume that the system obeys classical statistics, that is, that quantum statistical effects can be neglected. Therefore, the particle number density is

$$n(t, \mathbf{x}) = \int \frac{d^d \hat{p}}{h^d} f(t, \mathbf{x}, \hat{\mathbf{p}}). \tag{6}$$

We choose to divide the measure by a power of a constant $h$, with the dimensions of action, so that the integration measure above has units of inverse spatial volume, the same units as a particle number density. In this way, $f$ is properly dimensionless. We can also think of this as the analogue of the usual measure for the number of single-particle states within a phase space volume $d^d x \, d^d p$ (i.e., the density of states) in the context of quantum mechanics, which is $h^d$ according to the uncertainty principle were one to take $h$ to be the Planck constant. In the present context, we merely view $h$ as a dimensional bookkeeping factor of convenience.

By momentum, we specifically mean the appropriate component of the energy-momentum tensor associated with spatial translation-invariance:

$$\hat{p}_i \equiv T^0{}_i. \tag{7}$$

This identification is obvious in the boost-invariant case, but it will prove to be crucial in the absence of boost invariance.[5] In particular, we cannot posit an a-priori relationship between momentum $\hat{\mathbf{p}}$ and velocity,

$$\hat{\mathbf{v}} = \frac{d\mathbf{x}}{dt}, \tag{8}$$

such as the standard $\hat{\mathbf{p}} = m\hat{\mathbf{v}}$. Instead, one must define the kinetic mass density $\hat{\rho}$ in (3) such that $\hat{\mathbf{p}} = \frac{\hat{\rho}}{n}\hat{\mathbf{v}}$. The kinetic mass density is itself velocity-dependent and is thus not usually simply proportional to $n$. Galilean boost invariance, however, does set $\hat{\rho} = mn$ [1], in which case $\hat{\mathbf{p}} = \frac{\hat{\rho}}{n}\hat{\mathbf{v}}$ reduces to the usual $\hat{\mathbf{p}} = m\hat{\mathbf{v}}$. In the Lorentz boost-invariant case, we instead get $\hat{\mathbf{p}} = \hat{\gamma}m\hat{\mathbf{v}}$, where $\hat{\gamma} = (1 - \hat{v}^2)^{-1/2}$, so that[6] $\hat{\rho} = mn\hat{\gamma}$.

We denote the momentum-space expectation value of operators by dropping the hat, e.g.,

$$\mathbf{v} = \frac{1}{n} \int \frac{d^d \hat{p}}{h^d} f \, \hat{\mathbf{v}}. \tag{9}$$

We also use an overline to denote the momentum-space expectation value of more complicated composite operators, such as products of operators, e.g.,

$$\overline{\hat{v}^i \hat{v}^j} = \frac{1}{n} \int \frac{d^d \hat{p}}{h^d} f \, \hat{v}^i \hat{v}^j. \tag{10}$$

The distribution function satisfies the Boltzmann equation, which in its most basic form reads

$$\frac{df}{dt} = \left(\frac{\partial f}{\partial t}\right)_{\text{coll}}, \tag{11}$$

---

[5]Where, moreover, $T^0{}_i$ and $T^i{}_0$ require the specification of an additional potential in order to be related to each other [1] (see also Appendix A).

[6]Here we see that non-trivial velocity dependence of the kinetic mass density can also feature in relativistic boost invariant systems. However, for non-relativistic velocities $v/c \ll 1$, the usual Galilean relation results. In what follows, we allow for the dimensionful parameters that define the velocity dependence of kinetic mass density via (3) to be such that a non-negligible dependence can occur at any velocity.

where the left hand side is the total time derivative, and the right hand side is due solely to inter-particle collisions and interactions with the environment. As we discuss shortly, it is incumbent upon us to separately assess the implications of boost non-invariance for the different contributions to the collision term, in particular those corresponding to exchange with the environment.

This collision term is often denoted as a functional $C$ that depends on the distribution function $f$ and its equilibrium value $f^{\text{eq}}$:

$$\left(\frac{\partial f}{\partial t}\right)_{\text{coll}} = C(f, f^{\text{eq}}), \tag{12}$$

specified such that $C$ should tend to relax $f$ towards $f^{\text{eq}}$. For example, the BGKW model posits [18, 19]

$$C_{\text{BGKW}}(f, f^{\text{eq}}) = -\frac{f - f^{\text{eq}}}{\tau}, \tag{13}$$

where $\tau$ is some collision time (itself possibly space- and time-dependent).[7] This form for the collision term also goes by the name of the *single relaxation*, or *single collision time* approximation. Plugging this into the right hand side of (11) indeed shows that when $f > f^{\text{eq}}$, the evolution equation of $f$ will tend to decrease $f$, and if $f < f^{\text{eq}}$, it will tend to increase it, tending towards $f^{\text{eq}}$ in both cases. For $f$ sufficiently close to $f^{\text{eq}}$ this is simply linear response in the space of distributions. Of course, one could consider a more complicated dependence on $f - f^{\text{eq}}$ so long as the collision functional satisfies three requirements [37]:

1. *Detailed balance*: $C(f^{\text{eq}}, f^{\text{eq}}) = 0$, which means that when $f$ is at the equilibrium value, the rate of scattering out of the equilibrium state is perfectly balanced by the rate of scattering into the equilibrium state;

2. *Ward identities*: $\int d^d \hat{p} \, C(f, f^{\text{eq}}) \phi = 0$ for any conserved quantity $\phi$;

3. *H-theorem*: $\int d^d \hat{p} \, C(f, f^{\text{eq}}) H(f) \leq 0$ for any concave function $H$, such as $\log f$.

Detailed balance is guaranteed for any function with a well-defined Taylor series in $f - f^{\text{eq}}$, which is certainly the case for the BGKW ansatz. Although we won't make use of the *H*-theorem in what follows,[8] we will make use of the Ward identity mindfully, first observing that the momentum integral of the collision function itself should vanish because the function $\phi = 1$ is conserved. In the boost-invariant case, momentum conservation provides a separate Ward identity with $\phi \equiv \mathbf{p}$ for each component of the momentum. However, although this assumption continues to hold for the inter-particle contributions to the collision term due to the persistence of spatial translation invariance in the system, the same cannot be presumed for the environment-system contributions. This is due to the fact that momentum exchange can occur between the system and the reservoir. The consequences of the latter propagate through the usual derivation of the Vlasov hierarchy to eventually result in (4) in the absence of forces, and (5) in their presence, as opposed to the usual Euler and continuity equations that would have resulted in the boost-invariant case. We occupy ourselves with this derivation in the remainder of this section.

---

[7]The term "collision time" here refers to the time *between* collisions, as opposed to the duration of a collision. The time $\tau$ is sometimes also referred to as a "relaxation time". As usual, we assume that the duration of a collision can be ignored: that time scale is much shorter than $\tau$, the time between collisions. This assumption implicitly involves both density *and* interaction strength: the higher both of those are, the lower we expect $\tau$ to be, and the more one has to be careful about the approximation.

[8]The role of this requirement is to restrict the collision term to have a functional form such that the Gibbs entropy is non-decreasing. For eventual applications of the formalism presented here to active systems in complete generality, this condition would have to be revisited.

We begin by noting that the total time derivative of the distribution function can be written as

$$\frac{df}{dt} = \frac{\partial f}{\partial t} + \frac{dx^i}{dt}\frac{\partial f}{\partial x^i} + \frac{d\hat{p}_i}{dt}\frac{\partial f}{\partial \hat{p}_i}$$
$$= \frac{\partial f}{\partial t} + \hat{\mathbf{v}}\cdot\nabla f + \hat{\mathbf{F}}\cdot\nabla_{\hat{\mathbf{p}}}f, \tag{14}$$

where $\hat{\mathbf{F}}$ is the total force on the single particle at time $t$, at position $\mathbf{x}$, and with momentum $\hat{\mathbf{p}}$. Therefore, the Boltzmann equation (11) can be expressed as

$$\frac{\partial f}{\partial t} + \hat{\mathbf{v}}\cdot\nabla f + \hat{\mathbf{F}}\cdot\nabla_{\hat{\mathbf{p}}}f = C. \tag{15}$$

The Vlasov hierarchy arises from taking momentum expectation values of successive momentum moments of the above. For clarity of discussion, the derivation presented in this section will assume that there is no net force on the particles, in which case the Boltzmann equation simplifies to

$$\frac{\partial f}{\partial t} + \hat{\mathbf{v}}\cdot\nabla f = C. \tag{16}$$

The more general case incorporating forces (including momentum-dependent forces) is straightforward to follow once one is familiar with the force-free derivation. We present the details of the latter derivation in Appendices B and C, and will return to it in the next section when we discuss relating 'microphysical', kinetic theory-level descriptions of flocks (such as the Vicsek model [38]) to the 'macrophysical', hydrodynamic descriptions to which they flow under coarse graining [39].

## 2.1 Zeroth moment: Number density conservation

We first take the zeroth momentum moment of the forceless Boltzmann equation (16):

$$\frac{\partial}{\partial t}\underbrace{\int\frac{d^d\hat{p}}{h^d}f}_{=n} + \nabla\cdot\underbrace{\int\frac{d^d\hat{p}}{h^d}f\hat{\mathbf{v}}}_{=n\mathbf{v}} = \underbrace{\int\frac{d^d\hat{p}}{h^d}C}_{=0}. \tag{17}$$

The integral of $C$ vanishes because the quantity $\phi = 1$ is obviously conserved. This yields the equation for the conservation of particle number density in the absence of forces:

$$\partial_t n + \nabla\cdot(n\mathbf{v}) = 0. \tag{18}$$

As shown in equation (B.4), however, the number density $n$ is no longer conserved in the presence of generic momentum-dependent forces.

## 2.2 First moment: Pre-Euler equation

We now take the first momentum moment of (16):

$$\frac{\partial}{\partial t}\int\frac{d^d\hat{p}}{h^d}f\hat{p}^i + \partial_j\int\frac{d^d\hat{p}}{h^d}f\hat{p}^i\hat{v}^j = C^i, \tag{19}$$

where $C^i$ is defined as

$$C^i = \int\frac{d^d\hat{p}}{h^d}C\hat{p}^i. \tag{20}$$

We can write the above as

$$\partial_t(np^i) + \partial_j(n\overline{\hat{p}^i\hat{v}^j}) = C^i. \tag{21}$$

In order to express this as something resembling an Euler equation, we need to relate momenta to velocities. For this, we make use of the relation (3)

$$n\hat{\mathbf{p}} = \hat{\rho}\hat{\mathbf{v}},$$

so that equation (21) becomes

$$\partial_t\left(\overline{\hat{\rho}\hat{v}^i}\right) = -\partial_j\left(\overline{\hat{\rho}\hat{v}^i\hat{v}^j}\right) + C^i,\tag{22}$$

where $C^i$ can be expressed via (3) and (13) as:

$$C^i = -\frac{\rho}{\tau}\left(v^i - v^i_{\text{eq}}\right).\tag{23}$$

We now make the crucial assumption that we can factorize the expectation value of $\hat{\rho}$ from any other expectation value. This assumption derives from rotational-invariance and the additional assumption that we can ignore sufficiently high velocity moments.[9] Rotational-invariance implies that $\hat{\rho}$ depends on the quantity $\hat{v}^2$, rather than on $\hat{v}$ itself. Thus, if we can ignore third-order velocity moments, we can always factorize $\hat{\rho}$ out of any expectation value. Hence, equation (22) simplifies to

$$\partial_t(\rho v^i) = -\partial_j\left(\rho\,\overline{\hat{v}^i\hat{v}^j}\right) + C^i,\tag{24}$$

or

$$\partial_t v^i = -\frac{1}{\rho}\partial_j\left(\rho\,\overline{\hat{v}^i\hat{v}^j}\right) - \frac{v^i}{\rho}\partial_t\rho + \frac{C^i}{\rho}.\tag{25}$$

The modification to this relation in the presence of forces is given by (B.8).

In the boost-invariant context, momentum is conserved outright, and so the relevant Ward identity implies that $C^i$ is identically zero. In the present context, we can still appeal to the preservation of global translation symmetry for the system, which implies that the inter-particle or inter-boid interactions in a flock conserve momentum. Correspondingly, the contribution to the collision function that encodes inter-particle/boid collisions contributes vanishingly to $C^i$. However, the system-environment interactions do not conserve momentum for any particle/boid when considered in isolation, as is implicit in (22) and elaborated upon in the subsequent discussion, resulting in non-vanishing contributions to $C^i$.

## 2.3 Second moment: The generalized Euler equation

We now take the second momentum moment of (16):

$$\partial_t\left(n\overline{\hat{p}^i\hat{p}^j}\right) + \partial_k\left(n\overline{\hat{p}^i\hat{p}^j\hat{v}^k}\right) = C^{ij},\tag{26}$$

where $C^{ij}$ is the second momentum moment of the collision function. Within the approximations we have already made, the time-derivative term can be dropped. This follows from (18) and the assumption that we can neglect higher order velocity moments (cf. footnote 9), as well as the assumption that the total force on any particle/boid is vanishing. In relaxing the latter approximation as we do in Appendix B, we will make the additional assumption that the second and higher velocity moments relax sufficiently quickly due to collisions so that the

---

[9]This relates to the manner in which we truncate the Vlasov hierarchy that would ordinarily go on indefinitely by assuming the 'enslavement' of third- and higher-order moments. Generally, one assumes the enslavement of all moments of higher order than the one of interest. We are interested in the two-point function, and thus we will truncate at third order. This is the usual procedure (see [37]). We return to this point in Section 2.3.

time-derivative term can be dropped in the presence of forces as well. Next, we insert the BGKW collision functional, whose second momentum moment reads

$$C^{ij} = -\frac{n\overline{\hat{p}^i\hat{p}^j} - \left(n\overline{\hat{p}^i\hat{p}^j}\right)_{\text{eq}}}{\tau}.$$ 
(27)

Finally, we plug in the momentum-velocity relation (22) and again pull $\hat{\rho}$ out of any expectation value to obtain

$$\partial_k\left(\tfrac{1}{n}\rho^2\overline{\hat{v}^i\hat{v}^j\hat{v}^k}\right) = -\frac{\rho^2}{n}\frac{\overline{\hat{v}^i\hat{v}^j} - \left(\overline{\hat{v}^i\hat{v}^j}\right)_{\text{eq}}}{\tau}.$$ 
(28)

It is a common approach at this point to assume that the third and higher velocity moments are 'enslaved' to their equilibrium values. The assumption that we can pull $\hat{\rho}$ out of any expectation value relies on our ability to ignore third and higher velocity moments, which paraphrases the assumption of a hierarchy among interactions, where fast modes serve to factorize higher order correlations in terms of equilibrium values [37, 39]. To be consistent, therefore, we assume that we can neglect all such higher-order terms. Doing so has the corollary that the second velocity moment is indeed enslaved to its equilibrium value:

$$\overline{\hat{v}^i\hat{v}^j} = \left(\overline{\hat{v}^i\hat{v}^j}\right)_{\text{eq}}.$$ 
(29)

We then posit the following general ansatz for the equilibrium second velocity moment:

$$\left(\overline{\hat{v}^i\hat{v}^j}\right)_{\text{eq}} = v^i v^j + \frac{P}{\rho}\delta^{ij},$$ 
(30)

where $P$ is some function of $v^2$, which we would like to interpret as pressure, but which, for now, we take to be some arbitrary function. This is a standard expression used in the literature; it is the most general expression we can construct out of the expectation value $v^i$ that has the correct tensor structure and does not involve spatial gradients. The expression is understood to hold in the bulk of the fluid, where equilibrium conditions justify excluding spatial gradients. Of course, in the vicinity of boundaries spatial gradients may naturally arise and the situation will be more complicated. Therefore,

$$\rho\,\overline{\hat{v}^i\hat{v}^j} = \rho v^i v^j + P\delta^{ij}.$$ 
(31)

Plugging this into (25) and rearranging terms yields

$$\partial_t v^i + (\mathbf{v}\cdot\nabla)v^i = -\frac{\partial^i P}{\rho} - \frac{v^i}{\rho}\left[\partial_t\rho + \nabla\cdot(\rho\mathbf{v})\right] + \frac{C^i}{\rho}.$$ 
(32)

Having used the BGKW, or collision time form to express $C^{ij}$, we utilize the expression for $C^i$ (23) which results in:

$$\partial_t v^i + (\mathbf{v}\cdot\nabla)v^i = -\frac{\partial^i P}{\rho} - \frac{v^i}{\rho}\left[\partial_t\rho + \nabla\cdot(\rho\mathbf{v})\right] - \frac{v^i - v^i_{\text{eq}}}{\tau}.$$ 
(33)

Equation (1) as derived in [1] is the equilibrium limit of the above, where equilibrium corresponds to detailed balance with respect to momentum transfer between the system and the reservoir.[10] Before we are in a position to elaborate on the physical consequences of the modified Euler equation, it behooves us to understand how (33) gets modified in the presence of forces.

---

[10]In the context of flocks, for example, the last term in (33) enforces the boids to co-move with the ambient environment – the air through which birds fly, for example.

# 3 Incorporating inter-particle/boid forces

It is a straightforward, if slightly more laborious, exercise to repeat the preceding derivation in the presence of inter-particle/boid forces, as detailed in Appendix B. The net result is that instead of (33), one obtains (B.12):

$$
\begin{aligned}
\partial_t v^i + \left(\mathbf{v} \cdot \nabla\right) v^i = &-\frac{\partial^i P}{\rho} - \frac{v^i}{\rho}\left[\partial_t \rho + \nabla \cdot (\rho \mathbf{v})\right] - \frac{v^i - v_{\mathrm{eq}}^i}{\tau} \\
&+ \frac{nF^i}{\rho} - \frac{2}{\rho}\partial_j\left(\tau n \overline{\hat{v}^{(i}\hat{F}^{j)}}\right) + \overline{\hat{v}^i(\nabla_{\hat{\mathbf{p}}}\cdot\hat{\mathbf{F}})} - \frac{1}{\rho}\partial_j\left(\tau\rho\,\overline{\hat{v}^i\hat{v}^j(\nabla_{\hat{\mathbf{p}}}\cdot\hat{\mathbf{F}})}\right).
\end{aligned}
\tag{34}
$$

In order to proceed, one has to specify the forces $\hat{F}^i$ and subsequently perform the hydrodynamical averages in order to calculate the additional source terms in (34). We do this in Appendix C by positing a specific form for the inter-boid force component that can incorporate the Vicsek model with additional constraints, but is in principle more general. Specifically, we posit the simplest non-trivial parameterization of inter-boid forces to be

$$
\hat{\mathbf{F}} = -\lambda(\hat{\mathbf{v}} - \hat{\boldsymbol{\eta}}),
\tag{35}
$$

where $\lambda$ is some constant with dimensions of mass per time, and where $\hat{\boldsymbol{\eta}}$ is a local averaged velocity field with which the individual boids would tend to align in direction and in magnitude for positive values of $\lambda$. One might consider adding explicit noise terms to (35). However, under the usual assumptions, these will drop out of averaged expressions such as (34).

Furthermore, we note that unless we impose restrictions on the form of the distribution function, the parameterization (35) as expressed departs from anything that could reproduce the Vicsek model which considers normalized velocities, and consequently only aligns directions to a local averaged field (with window function-like support). The ansatz (35) instead allows for longitudinal breathing modes corresponding to boids or particles getting ahead of or catching up with the local flow. In order to reproduce anything that could resemble the Vicsek model, one would either have to restrict the distribution function to have no support outside a momentum shell that corresponds to the constraint $|\mathbf{v}| = 1$ (or some other constant), rendered non-trivial because of the relation (3). The net result would mimic adding additional contact force terms to (35) to enforce the momentum shell constraints. Equivalently, one can rederive the Vlasov hierarchy from the beginning but replacing the Poisson brackets implicit in the Boltzmann equation[11] (14) with the corresponding Dirac brackets, and considering the equivalent constrained Hamiltonian evolution [41–43]. We leave this as an exercise for the future, opting for the simple unconstrained parameterization (35) in what follows, as longitudinal breathing modes in a flock are evidently well motivated from physical considerations, and moreover, is straightforward to deal with in the calculations to follow. We could, of course, consider more general possibilities than the ansatz (35) as briefly discussed in Appendix C, however, this simple parameterization already possesses a rich phenomenology which warrants an expanded discussion. Of particular importance, is how microscopic stochasticity manifests at long wavelengths even as the addition of noise terms to (35) drops out of hydrodynamic averages.

We first emphasize that $\hat{\boldsymbol{\eta}}$ is not to be thought of as a (global) mean field. Instead, à la Vicsek, it is to be thought of as a *local average* of the velocities of the nearby boids whose precise form remains to be specified. It is essentially a non-local quantity, but in a manner that

---

[11]Where we note that in the case that inter-boid forces derive from an underlying symplectic structure, (14) can be rewritten as $\frac{df}{dt} = \frac{\partial f}{\partial t} + \frac{dx^i}{dt}\frac{\partial f}{\partial x^i} + \frac{d\hat{p}_i}{dt}\frac{\partial f}{\partial \hat{p}_i} \equiv \frac{\partial f}{\partial t} + \{\mathcal{H}, f\}$ where $\mathcal{H}$ is the generator of time evolution, and where $\{\,,\}$ denotes the corresponding Poisson bracket.

is made transparent and tractable when written, without loss of generality, as the convolution of an *influence kernel* $K^i{}_j(\mathbf{r}, \mathbf{r}')$ with the velocity field:

$$\hat{\eta}^i(\mathbf{r}) = \int d^d r' \, K^i{}_j(\mathbf{r}, \mathbf{r}') \, \hat{v}^j(\mathbf{r}') \,. \tag{36}$$

Isotropy dictates that the tensor structure of $K^i{}_j$ is proportional to the Kronecker delta, so that $K^i{}_j \propto \delta^i{}_j$, which, when supplemented with spatial translation invariance of the system, fixes its functional dependence to be of the form:

$$K^i{}_j(\mathbf{r}, \mathbf{r}') \equiv \delta^i{}_j K(|\mathbf{r} - \mathbf{r}'|) \,. \tag{37}$$

We note that the influence kernel will have finite support for short-range forces, and its precise form serves as a functional parameterization of all possible two-body inter-boid or inter-particle forces at the microscopic, or kinetic theory level.[12] The influence kernel captures only the inter-boid interactions, not the interactions between the boids and the environment. As stated earlier, we assume translation-invariance of the inter-boid interactions, which is why $K$ is a function only of $|\mathbf{r} - \mathbf{r}'|$. In addition, we will mostly assume that the inter-boid interactions are also isotropic, which is why $K^i{}_j \propto \delta^i{}_j$. Nevertheless, we show in Appendix C how anisotropic terms in the influence kernel can naturally lead to diffusion terms in the Euler equation.

A 'Vicsek-like' local averaged field would correspond to

$$K(|\mathbf{r} - \mathbf{r}'|) = \Theta(r_0 - |\mathbf{r} - \mathbf{r}'|) \,, \tag{38}$$

where $r_0$ is some radius around a given boid within which it can be influenced by its local flow, with equal weighting given to every other boid within this radius. Another plausible choice for a short-range influence kernel is given by:

$$K(|\mathbf{r} - \mathbf{r}'|) = \frac{\kappa}{4\pi} \frac{e^{-\mu|\mathbf{r} - \mathbf{r}'|}}{|\mathbf{r} - \mathbf{r}'|} \,, \tag{39}$$

which assigns decreasing influence with distance, and effectively screens all interactions beyond the length scale $r_0 = \mu^{-1}$. Although we consider various possibilities in two spatial dimensions in Appendix C, the formalism we present is straightforwardly implemented in an arbitrary number of dimensions. We use the form (39) for the purpose of illustrating certain generic features in the long wavelength limit, and because it has a particularly transparent interpretation in three spatial dimensions, where it happens to be the inverse of the Helmholtz (or screened Coulomb, or Yukawa) operator:

$$(\nabla^2 - \mu^2) K(|\mathbf{r} - \mathbf{r}'|) = -\kappa \delta^3(\mathbf{r} - \mathbf{r}') \,. \tag{40}$$

That is, inserting the forms (39) and (37) into (36), re-expressing $\mu$ as the inverse length scale $\mu = r_0^{-1}$, and restricting to wavelengths much bigger than $r_0$, so that $|r_0^2 \nabla^2| \ll 1$ for all modes of interest, one has the formally convergent operator identity:

$$\int d^d r' \, \delta^i{}_j K(|\mathbf{r} - \mathbf{r}'|) \hat{v}^j(\mathbf{r}') \equiv -\frac{\kappa}{\nabla^2 - \mu^2} \hat{v}^i(\mathbf{r}) = \kappa r_0^2 \left[ 1 + r_0^2 \nabla^2 + r_0^4 \nabla^4 + \dots \right] \hat{v}^i(\mathbf{r}) \,. \tag{41}$$

More generally, as we show in Appendix C, given an arbitrary influence kernel defined by a characteristic scale $r_0$, restricting to wavelengths much larger than $r_0$ will result in a derivative expansion for the local averaged field:

$$\hat{\eta}^i(\mathbf{r}) = \kappa r_0^2 \left[ 1 + \alpha_2 r_0^2 \nabla^2 + \alpha_4 r_0^4 \nabla^4 + \dots \right] \hat{v}^i(\mathbf{r}) \,, \tag{42}$$

---

[12]One can in principle include higher point interactions by adding additional quadratic and higher convolutions to (36) as desired, i.e.: $\hat{\eta}^i(\mathbf{r}) = \int d^d r' \, K^i{}_j(\mathbf{r}, \mathbf{r}') \hat{v}^j(\mathbf{r}') + \int d^d r' \int d^d r'' \, K^i{}_{jk}(\mathbf{r}, \mathbf{r}', \mathbf{r}'') \hat{v}^j(\mathbf{r}') v^k(\mathbf{r}'') + \dots$ etc.

where the specific coefficients $\alpha_i$ encapsulate the microscopic details of the inter-boid interactions,[13] and where the parameter $\kappa$ sets the amplitude of the underlying stochastic fluctuations were the phase space moments of the fluid derived from an underlying statistical field theory generating functional. This is a well-known procedure in the context of effective field theory in particle physics phenomenology (where $\hbar$ plays the role of $\kappa$), but is more broadly applicable in any statistical field-theoretic formalism. The weighted derivative expansion is how microscopic fluctuations – whatever their stochastic origin may be – communicate to long-wavelength observables.

How something that would ordinarily take the explicit addition of noise terms at the kinetic theory level [39] can be captured entirely from a simple derivative expansion might seem striking, so we illustrate how stochastic fluctuations of quantum-mechanical origin emerge as a derivative expansion with a simple example from electrostatics in Appendix D. Specifically, we illustrate how quantum-mechanical 'smearing' of sources (the origin of which are vacuum fluctuations) translates into higher-derivative corrections to the equations of motion, which will be derivative corrections to the Gauss law that relates electrostatic potentials to sources. Expressions such as (42) can be viewed as a transposition of this phenomenon to the present context if we were to view $\kappa$ as analogous to $\hbar$, or $\beta^{-1}$ in finite-temperature statistical field theory.

We note in concluding this section that although we have posited velocity-dependent forces in (35) for clarity of discussion, as stressed in Appendix C, it is more straightforward and entirely equivalent to posit momentum-dependent forces and rely upon (3) to relate the two to each other. Furthermore, we note that there is nothing stopping us from relaxing the assumptions regarding the order of velocity moments that we keep in our approximation if one is interested in computing higher-order corrections to (33) and (34). Doing so would have simply resulted in more complicated expressions that are to be understood in terms of a phenomenological power-counting, or derivative, expansion,[14] of which equations (33) and (34) represent the leading terms.

## 4 The hydrodynamics of flocks

We would like to compare the Euler equation in the absence of forces (33) to the analogous Euler equation of Toner-Tu theory as summarized in [22], which reads

$$\partial_t \mathbf{v} + \lambda_1 (\mathbf{v} \cdot \nabla) \mathbf{v} + \lambda_2 (\nabla \cdot \mathbf{v}) \mathbf{v} + \lambda_3 \nabla (v^2) = (\alpha - \beta v^2) \mathbf{v} - \frac{\nabla P_0}{\rho_0} \,, \tag{43}$$

where we drop diffusion terms for the time being.[15] We subscript the density and pressure referenced in (43) with zeroes to distinguish them from the corresponding quantities in the present treatment, since these are not the same: in [22], these quantities are taken to be independent of velocity, which, as discussed in previous sections, is not something one can take for granted on general kinematic grounds for a boost non-invariant fluid.

In order to proceed, we introduce the notion of an equation of kinetic state, which can be viewed as a constitutive relation whose specific form characterizes the fluid at hand that

---

[13]Specifically, different functional forms of the influence kernel in various dimensions can only appear as different coefficients in the expansion (42) at long wavelengths.

[14]Or, in terms of a relevant/ marginal/ irrelevant operator expansion in renormalization group terminology.

[15]Implicit in this equation is the assumption of the analyticity of the small-$\mathbf{v}$ expansion as, otherwise, scalar quantities in this equation could have depended on $v = |\mathbf{v}|$ instead of $v^2 = |\mathbf{v}|^2$. We assume this throughout as well.

parameterizes the non-conservation of the kinetic mass density:[16]

$$\partial_t \rho + \nabla \cdot (\rho \mathbf{v}) = g \rho, \tag{44}$$

where $g$ (the equation of kinetic state) is a scalar function that can be built out of the velocity. In fact, it need not be a function solely of $v^2$ (assuming isotropy), but also of $\nabla \cdot \mathbf{v}$ and more generally, a whole hierarchy of terms consistent with the residual symmetries of the system, such as $\nabla_i \nabla_j v^i v^j$, and so on. However, power counting velocity at the same order as a spatial derivative and truncating our analysis to second-order implies that $g$ can be specified by just three coefficients, which we choose with foresight as $\alpha$, $\beta$, and $\lambda_2$:

$$g(\nabla \cdot \mathbf{v}, v^2) = -\alpha + \beta v^2 + \lambda_2 \nabla \cdot \mathbf{v}. \tag{45}$$

Recalling (33)

$$\partial_t v^i + (\mathbf{v} \cdot \nabla) v^i = -\frac{\partial^i P}{\rho} - \frac{v^i}{\rho} \left[ \partial_t \rho + \nabla \cdot (\rho \mathbf{v}) \right] - \frac{v^i - v^i_{\text{eq}}}{\tau},$$

and inserting the parameterization (45) in place of the term in the square parenthesis results in the intermediate expression:

$$\partial_t \mathbf{v} + (\mathbf{v} \cdot \nabla) \mathbf{v} + \lambda_2 (\nabla \cdot \mathbf{v}) \mathbf{v} = (\alpha - \beta v^2) \mathbf{v} - \frac{\nabla P}{\rho} - \frac{\mathbf{v} - \mathbf{v}_{\text{eq}}}{\tau}. \tag{46}$$

We now make the further assumption that the system is barotropic. This means that the pressure only depends on space and time via its implicit dependence on $\rho$ and the generalized chemical potentials to which it relates via a conventional equation of state. This equation of state can also depend on $v^2$ up to quadratic order in velocities, but one could also consider dependence on temperature and other potentials. Furthermore, we suppose that the pressure $P_0$ which appears in (43) is obtained form that part of the barotropic relationship which does not depend on $v^2$, and that we can expand this relationship in powers of $v^2$:

$$P = P(\rho, v^2). \tag{47}$$

This thus means that the quantities $P_0$ and $\rho_0$ are given by

$$P_0 = P(\rho_0, 0), \quad \rho_0 = \rho \big|_{v^2 = 0}. \tag{48}$$

We can exclude any explicit dependence in (47) on $\nabla \cdot \mathbf{v}$ because that is not the chemical potential for any quantity, where we recall that the Maxwell relations between the thermodynamic potentials are derived by Legendre transformation with respect to the chemical potentials. As a result, one finds that

$$-\frac{\nabla P}{\rho} = -\left( \frac{\partial P}{\partial \rho} \right)_{v^2} \frac{\nabla \rho}{\rho} - \frac{1}{\rho} \left( \frac{\partial P}{\partial (v^2)} \right)_{\rho} \nabla(v^2), \tag{49}$$

implying that equation (46) can be expressed as

$$\partial_t \mathbf{v} + (\mathbf{v} \cdot \nabla) \mathbf{v} + \lambda_2 (\nabla \cdot \mathbf{v}) \mathbf{v} + \frac{1}{\rho} \left( \frac{\partial P}{\partial (v^2)} \right)_{\rho} \nabla(v^2) = (\alpha - \beta v^2) \mathbf{v} - \left( \frac{\partial P}{\partial \rho} \right)_{v^2} \frac{\nabla \rho}{\rho} - \frac{\mathbf{v} - \mathbf{v}_{\text{eq}}}{\tau}. \tag{50}$$

---

[16]We recall that although the zeroth term of the Vlasov hierarchy ensures the conservation of number density (18), the non-trivial nature of the relation (3) does not imply the conservation of the kinetic mass density as a corollary.

Comparison with the analogous quantities from the Toner-Tu expression (43) leads us to the following identifications:

$$\lambda_1 = 1\,, \qquad \lambda_2 = \frac{\partial g}{\partial (\nabla \cdot \mathbf{v})}\bigg|_{\nabla \cdot \mathbf{v} = v^2 = 0}\,, \qquad \lambda_3 = \frac{1}{\rho}\left(\frac{\partial P}{\partial (v^2)}\right)_\rho\bigg|_{v^2 = 0}\,, \qquad (51)$$

$$-\alpha = g|_{\nabla \cdot \mathbf{v} = v^2 = 0}\,, \qquad \beta = \frac{\partial g}{\partial v^2}\bigg|_{\nabla \cdot \mathbf{v} = v^2 = 0}\,, \qquad \nabla P_0 = \left[\left(\frac{\partial P}{\partial \rho}\right)_{v^2}\nabla \rho\right]\bigg|_{v^2 = 0}\,. \qquad (52)$$

We stress that these identifications are at leading order in the hydrodynamic expansion. The parameters $\lambda_1$, $\lambda_3$, and the pressure gradient term can all get corrections beyond quadratic order in velocity corrections to (30), which was the ansatz we assumed for the expectation value of the tensor product of two factors of the velocity, as well as additional terms in the Euler equation that arise from incorporating higher-order velocity moments as per the discussion below (22).

Before we consider the effect of incorporating forces, it is useful to examine the relationship and hierarchy between different coefficients that follow from kinematics and symmetries alone. We first note that the equation of kinetic state $g(\nabla \cdot \mathbf{v}, v^2)$, as we've parameterized it, must vanish in the boost-invariant limit. This follows simply from the defining equation (44), and the proportionality of the kinetic mass density $\rho$ to the number density in this limit via the relation $\rho = mn$ or $\rho = \gamma mn$, where $n$ is the number density, and where the latter is a conserved quantity via the zeroth-order Vlasov equation (18). From (45) we see that this implies that $\alpha, \beta$, and $\lambda_2$ represent parameterized departures from the boost-invariant limit, and must all vanish when boost invariance is restored. Similarly, we note from (51) that $\lambda_3$ also must vanish in the boost-invariant limit, as the dependence of the pressure on the velocity potential must vanish in this case.[17]

None of this should come as a particular surprise, as all one needs to do is to compare (50) to the boost-invariant Euler equation to infer what the correct limits have to be. What is non-trivial, however, is the fact that one can expect a hierarchy in the coefficients of the Toner-Tu description of active flocks on kinematic grounds alone, specifically:

$$\begin{aligned} \lambda_1 &= 1 + \mathcal{O}(v^3)\,, \\ \lambda_2 &\approx 0 + \mathcal{O}(v^3)\,, \end{aligned} \qquad (53)$$

where the additional terms are beyond quadratic order in the velocities that we have systematically neglected, not least through our truncation of the Vlasov hierarchy as discussed in Section 2. That is, the coefficient $\lambda_1$ tends to be close to unity, and the coefficient $\lambda_2$ tends to vanish. The reason for the latter fact can be seen via two arguments. Firstly, on general grounds, one can expect from the constitutive nature of the equation of kinetic state that its dependence to leading order will be on the velocity potential itself and not its derivatives,[18] implying a small to vanishing $\lambda_2$ via (51). Secondly, as we elaborate in our discussion of the physical content of the equation of kinetic state in Appendix E, the vanishing of $\lambda_2$ to leading order at long wavelengths can be understood in terms of power counting relevance for any operator expansion of the system/ environment interactions.

In the context of particle physics effective field theories, the tendency for dimensionless parameters to either vanish or be close to unity in certain units (being exactly zero or unity when

---

[17]We note a similarly spirited derivation in [44] which also took the boost-non-invariant hydrodynamics of [1] as starting vantage. The differing conclusions for the coefficients in (51) and (52) can be traced to including terms in the equation of kinetic state corresponding to convective derivatives of $v$ in the equation of kinetic state that are strictly subleading for us (as argued in Appendix E), as well as the distinctions made between (47) and (48).

[18]Furthermore, any subleading dependence on derivatives has to be consistent with isotropy, and so can only depend on the combination $\nabla \cdot \mathbf{v}$ which can only be non-trivial if the velocity potential field, and by extension, the equation of kinetic state is somewhere singular.

Table 1: A selection of isotropic influence kernels in 2d, their Fourier transforms, and the corresponding diffusion coefficient $D_T$ ($\nabla^2 \mathbf{v}$ term).

| Name | $\frac{1}{\kappa}K(r/r_0)$ | $\frac{1}{\kappa}\widetilde{K}(r_0 k)$ | $D_T$ (units of $\lambda\kappa r_0^2$) |
|---|---|---|---|
| Sharp cut-off | $\frac{1}{\pi r_0^2}\Theta(1 - \frac{r}{r_0})$ | $\frac{2}{r_0 k}J_1(r_0 k)$ | $\frac{1}{8}$ |
| Exponential | $\frac{1}{2\pi r_0^2}e^{-r/r_0}$ | $\left[1 + (r_0 k)^2\right]^{-3/2}$ | $\frac{3}{2}$ |
| Gaussian | $\frac{1}{2\pi r_0^2}e^{-\frac{1}{2}(r/r_0)^2}$ | $e^{-\frac{1}{2}(r_0 k)^2}$ | $\frac{1}{2}$ |

a symmetry is restored) is known as *technical naturalness* [45]. It is a statement about how hierarchies are preserved under renormalization group flow at long wavelengths, and boils down to the fact that these parameters are multiplicatively (rather than additively) renormalized as one flows to the infrared.[19] Whether the notion of technical naturalness transposes to the present context to yield the hierarchy (53) from a renormalization group analysis as suggested by our treatment in Appendix E is an important question we leave for the future. In this regard, we should note that the question of a proper and complete renormalization group analysis for the theory of flocks is not settled (see, for example, [46–49]). For the present, we note that under a minimal set of assumptions, a prediction of our treatment is that any active flock that satisfies the assumption of isotropy and inter-boid momentum conservation will satisfy the relations (53).

In order to see the effect of incorporating forces, we consider the result of inserting (36) along with the force ansatz (35) into (34) and subsequently performing the averages as detailed in Appendix C, with the net result that (50) gets modified to (C.31):

$$\partial_t \mathbf{v} + \lambda_1(\mathbf{v}\cdot\nabla)\mathbf{v} + \lambda_2(\nabla\cdot\mathbf{v})\mathbf{v} + \lambda_3\nabla(v^2) = (\alpha - \beta v^2)\mathbf{v} - \frac{\nabla P_0}{\rho_0} - \frac{\mathbf{v} - \mathbf{v}_{\text{eq}}}{\tau} \\ + D_T\nabla^2\mathbf{v} + D_B\nabla(\nabla\cdot\mathbf{v}), \tag{54}$$

with the same identifications made in (51) and (52), except for $\alpha$, which is modified according to

$$\alpha = -g\Big|_{\nabla\cdot\mathbf{v} = v^2 = 0} - \frac{d+1}{\tau'}, \tag{55}$$

where $\tau'$ is a characteristic time scale for a drag force that is linear in momentum.

The diffusion coefficients themselves are given by the parameter $\lambda$ that features in the force ansatz (C.1) and the parameters that define the functional form of the influence kernel, as illustrated in Table 1 for isotropic kernels (sampled from Table 2 in Appendix C). A non-zero $D_B$ diffusion coefficient can be generated through an anisotropic influence kernel (C.20), where furthermore, $D_B$ and $D_T$ relate to each other via the parameterized departure from isotropy. Finally, noise can be added in by hand, as is done in Toner-Tu theory.

## 5 Discussion and outlook

Hydrodynamic descriptions derive from an underlying kinetic theory in the long wavelength limit, and inherit the kinematics and dynamics governing the microscopic constituents through

---

[19]That is, the renormalization group equations for a given parameter is proportional to powers of the difference of that parameter from its symmetry restored value at any given renormalization group scale.

the Boltzmann equation and collision terms. If this kinematics is boost-invariant, then so is the hydrodynamic limit unless one introduces mechanisms to spontaneously break this invariance via the dynamics of the microscopic constituents. The manner in which the hydrodynamic description departs from the boost invariance will therefore be reflective of the details of this dynamics. On the other hand, if the underlying kinematics explicitly breaks boost-invariance, then not only should one expect this to transmit to the hydrodynamic limit, one might expect the resulting expansion to inherit a structure determined by the kinematics alone. Our investigation has confirmed that this is indeed the case.

Our primary motivations were two-fold. We firstly wanted to demonstrate a kinetic theory derivation of the boost-non-invariant hydrodynamics of [1] from a minimal set of assumptions consistent with the absence of boost-invariance. Secondly, given that any formalism that purports to generality should reproduce hydrodynamic limits of any system that lacks boost symmetries, we set about applying our findings to the hydrodynamics of flocks [20–22]. We arrived at (54) as the hydrodynamic limit that follows from our truncation to second order velocity moments, where the various coefficients of the expansion are determined by the equation of kinetic state (44) and (45), the velocity potential dependence of the pressure, the collision time corresponding to system/ reservoir momentum exchange, and the functional form of the influence kernel that inherits the microscopic details of the inter-boid interactions.

The main insights that led us to our conclusions, from which our derivations depart ab initio from standard treatments (see e.g. [26, 39]) involve:

- Applying an ontological distinction between the kinetic mass density, $\rho$, and the quantity $mn$, where $n$ is the particle number density, and

- acknowledging that in the absence of boost invariance, one must necessarily factor in the possibility of momentum exchange with the reservoir.

The key point is that when particles can exchange momentum with the reservoir, it is the quantity $\rho$ (3), and not $mn$ which measures how easy or hard it is to accelerate those particles with a fixed applied force (cf. appendix E). This forces relations between some of the coefficients of Toner-Tu theory [20–22] given that they derive from the equation of kinetic state defined in (44), which measures the degree of non-conservation of the kinetic mass density. When considering forces, we proceeded systematically via the introduction of an influence kernel, and utilized a standard technique in particle physics effective field theory to perform a derivative expansion of the kernel, obtaining different values for the coefficients of the leading operators in Toner-Tu theory as the long wavelength manifestation of different models of the microscopic interactions. An immediate extension of our formalism would be to consider retarded kernels in order to study propagation and additional transport effects in flocks, which we leave for a future study.

It is informative to compare our results to the process of course-graining a specific microscopic model with the standard Newtonian (i.e. Galilean boost invariant) kinematic relation between $\rho$ and $mn$, as performed in [39]. Here, one can derive the Dean equation from the Langevin equations describing the microscopic interactions, and systematically expand for the leading order modes at long wavelengths. The Toner-Tu theory which the Vicsek model flows to corresponds to a description where the following ratios are fixed as: $\lambda_1/\lambda_2 = 3/5$ and $\lambda_2/\lambda_3 = -2$. In [40], one can also find a derivation of Toner-Tu theory from a coarse-graining of the Vicsek model via a different approach, and it would be useful to compare the two methods to each other. However, the comparison would be tangential to the present approach as both start from a specific model rather than a generic model parameterization, and differ in assuming the usual Newtonian relation between momentum density, number density, particle mass, and velocity. In both of the aforementioned approaches, among other things, there is no scope for $\lambda_2$ to vanish, which is the condition assumed in [22] in order to derive relationships

between scaling exponents. In our approach, on the other hand, $\lambda_2 = 0$ simply translates to $g$ being independent of $\nabla \cdot \mathbf{v}$ at leading order, albeit in a model which qualitatively differs from the Vicsek model in that it allows for longitudinal breathing modes via the parameterizations (35) and (36). Whether or not $\lambda_2 = 0$ also follows from a renormalization group analysis within our formalism applied to an influence kernel specific to the Vicsek model is an important question which behooves us to revisit.

We remark in closing that although our introduction only mentioned fluids that lack Galilean or Lorentzian boosts as examples of boost non-invariant systems, there is in fact a third possible type of boost symmetry – Carrollian symmetry – arising from the small speed of light limit of a Lorentz boost.[20] The thermodynamics and hydrodynamics of systems with Carroll boost symmetries have only been studied fairly recently [1, 50–54]. This is thus yet another example in which one can have breaking of boost symmetry. Relatedly, the hydrodynamic description of fractons [55–59] is an example of a non-boost invariant fluid with additional low-energy Goldstone modes due to spontaneous breaking of the dipole symmetry. This dipole symmetry behaves in a manner similar to Carroll boosts, and it would be worthwhile to examine whether the kinetic approach of this article can further inform the latter studies. We furthermore note that Lifshitz hydrodynamics (which also lacks boost invariance) has also been examined from a holographic perspective [60–65], in which there is a dual gravitational description in terms of moving black branes in a higher-dimensional spacetime. Whether some of the results obtained in the present investigation have dual formulations would also be an interesting avenue to pursue (cf. [66] for a review on Lifshitz holography).

# Acknowledgments

We wish to thank Jay Armas, Luca Giomi, Emil Have, Silke Henkes, Wim van Saarloos, and Koenraad Schalm for valuable discussions and comments on the manuscript. N.O. thanks the Instituut-Lorentz for Theoretical Physics in Leiden for hospitality. In addition, K.G. and S.P. wish to thank Nordita for hospitality when this work was initiated.

**Funding information** The work of N.O. is supported in part by VR project Grant 2021-04013 and Villum Foundation Experiment Project No. 00050317.

# A   Perfect fluids without boost invariance – A brief recap

In this appendix we briefly review hydrodynamics of systems that do not necessarily have a boost symmetry, focussing on the perfect fluid description as presented in Ref. [1].[21] Such systems thus include general non-boost invariant systems, while the formulation includes fluids with Lorentz, Galilean or Carrollian boost symmetry as a special cases. It also includes Lifshitz fluids, in case there is a scaling symmetry characterized by a critical exponent $z$, distinguishing between time and space. See Refs. [13–16] for further developments, such as the computation of first-order transport coefficients, the formulation on curved spacetime and the Schwinger-Keldysh approach.

In the following we assume only time and space translational symmetries along with rotational symmetries, generated by $H$, $P_i$ and $J_{ij}$ respectively. We furthermore assume that there is a global $U(1)$ symmetry generated by a charge $Q$, corresponding to electric charge or

---

[20]For an introduction to Carrollian symmetries and theories see e.g. [50].

[21]Certain aspects of hydrodynamics without boost symmetry were studied earlier, see e.g. [32] and references therein.

particle number, for example. These symmetries follow in the usual way from the conserved energy-momentum tensor $T^{\mu}{}_{\nu}$ and the $U(1)$ current $J^{\mu}$ with $\mu = (0, i)$, $i = 1 \ldots d$ in $D = d + 1$ dimensions.

A general consequence of the absence of boost symmetry is that different inertial frames are no longer related by boost transformations. This means that the fluid description has to include velocity $\mathbf{v}$ as an additional thermodynamic potential. The reason for this is as simple as its ramifications are profound, and can be arrived at directly by re-examining the construction of the canonical ensemble in the presence of boost invariance, which we take to be Lorentz boosts for concreteness.[22] That one obtains inverse temperature as the potential corresponding to energy exchange with the reservoir is a statement in a very specific frame – energy itself is not a relativistic scalar, rather the time-like component of an energy-momentum four vector. By boosting to a moving frame, one cannot avoid exchanging momentum with the reservoir as well. Therefore, one should have always been considering the density matrix constructed from the relevant Boltzmann weight:

$$\rho_{\beta} = \frac{e^{-\beta^{\mu} P_{\mu}}}{\mathcal{Z}} = \frac{e^{-\beta(H - \mathbf{v} \cdot \mathbf{P})}}{\mathcal{Z}}, \tag{A.1}$$

where we've parameterized the components of $\beta^{\mu}$ as $\beta^{\mu} = (\beta, -\beta \mathbf{v})$, where $v^i$ corresponds to the derivative of the logarithm of the density of states with respect the corresponding spatial component of relativistic energy-momentum vector. In the boost invariant context, one can always boost to a frame where $\beta^{\mu}$ has no spatial components, reproducing the standard Boltzmann density matrix. However, in the absence of boost invariance, one can no longer presume this is possible, and so one must retain (A.1) and its generalization to the grand canonical ensemble (2) as the most general possibility.

Furthermore, the absence of a boost Ward identity[23] leads to a new fluid variable, called the kinetic mass density $\rho$. In particular, the first law of thermodynamics can be written as

$$dP = s\, dT + n\, d\mu + \frac{1}{2} \rho\, dv^2, \tag{A.2}$$

when viewing the pressure $P(T, \mu, v^2)$ as a function of the temperature $T$, chemical potential $\mu$ and velocity $v$. Here $s$ is the entropy density and $n$ the $U(1)$ charge density. Thus, the kinetic mass density can be computed as

$$\rho(T, \mu, v^2) = 2 \left( \frac{\partial P}{\partial v^2} \right)_{T, \mu}. \tag{A.3}$$

It can be shown that for standard Galilean fluids this quantity reduces to a constant $\rho = mn$, while in the Lorentzian case it is equal to the sum of energy and pressure, implying a specific dependence on the velocity governed by the Lorentz factor [1]. In general it can be an arbitrary function of $v^2$, something that plays a central role in the main body of this paper.

It follows from the symmetries mentioned above, along with the condition of thermodynamic equilibrium, that a perfect fluid in the LAB frame has energy-momentum tensor and $U(1)$ current of the form

$$T^0{}_0 = -\mathcal{E}, \qquad T^0{}_j = \mathcal{P}_j, \qquad T^i{}_0 = -(\mathcal{E} + P) v^i, \qquad T^i{}_j = P \delta^i_j + v^i \mathcal{P}_j, \tag{A.4}$$

$$J^0 = n, \qquad J^i = n v^i. \tag{A.5}$$

---

[22]For a classification of all possible representations of phases of matter that result from spontaneously breaking Lorentz boost invariance, see [67].

[23]For massive Galilean systems, boost symmetry implies that the momentum current is equal to the mass current. For systems with Lorentz invariance the corresponding Ward identity says that the energy and momentum current are equal. Finally, Carroll boost symmetry implies that the energy current vanishes.

Here $\mathcal{E}$ is the energy density and $\mathcal{P}_i = \rho v_i$ the momentum density. The generalized Euler equation can then be derived from the conservation equations $\partial_\mu T^\mu{}_\nu = \partial_\mu J^\mu = 0$. It takes the form

$$\partial_0 \mathbf{v} + (\mathbf{v} \cdot \nabla)\mathbf{v} = -\frac{1}{\rho}\nabla P - \frac{\mathbf{v}}{\rho}\Big[\partial_0 \rho + \partial_i(\rho v^i)\Big]. \tag{A.6}$$

For a Galilei fluid when $\rho = mn$ is the mass density, the last term vanishes and the equation reduces to the well-known (sourceless) Euler equation. Finally, we note that one can obtain the following conserved entropy current

$$\partial_0 s + \partial_i\left(s v^i\right) = 0, \tag{A.7}$$

so that indeed perfect fluids have no entropy production and are thus non-dissipative.

# B  Vlasov hierarchy with forces

For convenience, we rewrite here the Boltzmann equation in the presence of forces:

$$\frac{df}{dt} = \frac{\partial f}{\partial t} + \hat{\mathbf{v}} \cdot \nabla f + \hat{\mathbf{F}} \cdot \nabla_{\hat{\mathbf{p}}} f. \tag{B.1}$$

Integrating this equation to take the zeroth momentum moment gives

$$\frac{\partial}{\partial t}\underbrace{\int \frac{d^d\hat{p}}{h^d} f}_{=n} + \nabla \cdot \underbrace{\int \frac{d^d\hat{p}}{h^d} f\hat{\mathbf{v}}}_{=n\mathbf{v}} + \int \frac{d^d\hat{p}}{h^d}\hat{\mathbf{F}} \cdot \nabla_{\hat{\mathbf{p}}} f = \underbrace{\int \frac{d^d\hat{p}}{h^d} C}_{=0}. \tag{B.2}$$

Let us assume that the momentum derivative can be integrated by parts with no associated boundary terms.[24] Thus,

$$\partial_t n + \nabla \cdot (n\mathbf{v}) = n\overline{\nabla_{\hat{\mathbf{p}}} \cdot \hat{\mathbf{F}}}, \tag{B.3}$$

where

$$\overline{\nabla_{\hat{\mathbf{p}}} \cdot \hat{\mathbf{F}}} = \frac{1}{n}\int \frac{d^d\hat{p}}{h^d} f \, \nabla_{\hat{\mathbf{p}}} \cdot \hat{\mathbf{F}}. \tag{B.4}$$

We see that we would have the usual continuity equation for the particle number density were it not for momentum-dependent forces. For example, a drag force that is linear in momentum would produce a source term proportional to $n$ on the RHS of the equation (B.3).

The first momentum moment of (B.1) reads

$$\frac{\partial}{\partial t}\int \frac{d^d\hat{p}}{h^d} f\hat{p}^i + \partial_j \int \frac{d^d\hat{p}}{h^d} f\hat{p}^i\hat{v}^j + \int \frac{d^d\hat{p}}{h^d}\hat{p}^i\hat{F}^j\partial_{\hat{p}^j} f = C^i, \tag{B.5}$$

where $C^i$ is the first momentum moment of $C$.

Integrating by parts in momentum space where necessary, we find

$$\partial_t(np^i) + \partial_j(n\overline{\hat{p}^i\hat{v}^j}) - n\overline{\partial_{\hat{p}^j}(\hat{p}^i\hat{F}^j)} = C^i, \tag{B.6}$$

which we write as

$$\partial_t(np^i) = -\partial_j(n\overline{\hat{p}^i\hat{v}^j}) + C^i + nF^i + n\overline{\hat{p}^i(\nabla_{\hat{\mathbf{p}}} \cdot \hat{\mathbf{F}})}. \tag{B.7}$$

---

[24]Ordinarily, this is a fine assumption as we expect $f$ to decay exponentially with the magnitude of the momentum. This decay overwhelms any polynomial momentum-growth in the force, as one might expect from drag, for example.

Using the relationship $n\hat{\mathbf{p}} = \hat{\rho}\hat{\mathbf{v}}$ and the assumption that we can move $\hat{\rho}$ out of any expectation value, we ultimately derive the equation

$$\partial_t v^i = -\frac{1}{\rho}\partial_j(\rho\,\overline{\hat{v}^i\hat{v}^j}) - \frac{v^i}{\rho}\partial_t\rho + \frac{C^i}{\rho} + \frac{nF^i}{\rho} + \overline{\hat{v}^i\big(\nabla_{\hat{\mathbf{p}}}\cdot\hat{\mathbf{F}}\big)}. \tag{B.8}$$

Take the second momentum moment of (B.1):

$$\partial_t(n\overline{\hat{p}^i\hat{p}^j}) + \partial_k(n\overline{\hat{p}^i\hat{p}^j\hat{v}^k}) - 2n\overline{\hat{p}^{(i}\hat{F}^{j)}} - n\,\overline{\hat{p}^i\hat{p}^j(\nabla_{\hat{\mathbf{p}}}\cdot\hat{\mathbf{F}})} = C^{ij}. \tag{B.9}$$

As discussed below (26), we assume that we can also drop the time-derivative term in the presence of forces. We plug in the BGKW collision function and the momentum-velocity relation (22). Again, we assume that we can pull $\hat{\rho}$ out of expectation values. Finally, we drop third and higher velocity moments. The resulting equation may be written in the form

$$\rho\,\overline{\hat{v}^i\hat{v}^j} = \rho\left(\overline{\hat{v}^i\hat{v}^j}\right)_{\text{eq}} + 2\tau n\,\overline{\hat{v}^{(i}\hat{F}^{j)}} + \tau\rho\,\overline{\hat{v}^i\hat{v}^j\big(\nabla_{\hat{\mathbf{p}}}\cdot\hat{\mathbf{F}}\big)}. \tag{B.10}$$

We posit the same ansatz for the equilibrium second moment as in equation (30):

$$\rho\,\overline{\hat{v}^i\hat{v}^j} = \rho v^i v^j + P\delta^{ij} + 2\tau n\overline{\hat{v}^{(i}\hat{F}^{j)}} + \tau\rho\,\overline{\hat{v}^i\hat{v}^j\big(\nabla_{\hat{\mathbf{p}}}\cdot\hat{\mathbf{F}}\big)}. \tag{B.11}$$

Plugging this into (B.8), using the BGKW form for $C^i$ in equation (23), and rearranging terms gives

$$\begin{aligned}
\partial_t v^i + \big(\mathbf{v}\cdot\nabla\big)v^i = &-\frac{\partial^i P}{\rho} - \frac{v^i}{\rho}\big[\partial_t\rho + \nabla\cdot(\rho\mathbf{v})\big] - \frac{v^i - v_{\text{eq}}^i}{\tau} \\
&+ \frac{nF^i}{\rho} - \frac{2}{\rho}\partial_j\big(\tau n\overline{\hat{v}^{(i}\hat{F}^{j)}}\big) + \overline{\hat{v}^i\big(\nabla_{\hat{\mathbf{p}}}\cdot\hat{\mathbf{F}}\big)} - \frac{1}{\rho}\partial_j\big(\tau\rho\,\overline{\hat{v}^i\hat{v}^j\big(\nabla_{\hat{\mathbf{p}}}\cdot\hat{\mathbf{F}}\big)}\big).
\end{aligned} \tag{B.12}$$

In the following section, we will consider a fairly simple ansatz for the force from which we will be able to reproduce the remaining terms in the Toner-Tu equation.

## C  The influence kernel and local mean-field velocity matching

For an *isotropic* flock, we cannot introduce a term, $\hat{\mathbf{F}}_0$, in the force which is independent of momentum or velocity, even if it were a function of space and time, *unless* it takes one or more of the following forms:

1. $\hat{\mathbf{F}}_0$ is a *conservative force* and is therefore the gradient of a potential, $\hat{\mathbf{F}}_0 = -\nabla\hat{U}$; or

2. $\hat{\mathbf{F}}_0$ is a *random force* with an isotropic distribution.

Case 1. is already contained in the pressure term. Case 2. corresponds to the random force term in the full Toner-Tu equation.[25] Therefore, in the following, we will ignore such a term and start with the linear ansatz

$$\hat{\mathbf{F}} = -\lambda(\hat{\mathbf{p}} - \hat{\boldsymbol{\eta}}), \tag{C.1}$$

where $\lambda$ is a constant with dimensions of inverse time and $\boldsymbol{\eta}$ is a "local mean-field momentum" term, which is a kind of *average* of the momenta of the nearby boids. With $\lambda > 0$, this force tends to align and match an individual boid's momentum with the local mean-field momentum

---

[25]Note that, in our case, the random force is multiplied by $\frac{n}{\rho}$, which is correct because the left-hand side of the Euler equation is an acceleration: the random force in [22] should more properly be called a random acceleration.

of the nearby boids. We write the local mean-field momentum as the convolution of a spatially-averaging *kernel* $K^i{}_j(\mathbf{r})$ with the momentum field:

$$\hat{\eta}^i(\mathbf{r}) = \int d^d r' \, K^i{}_j(\mathbf{r} - \mathbf{r}') \hat{\mathbf{p}}(\mathbf{r}'), \tag{C.2}$$

where, $\hat{\mathbf{p}}$ is technically not a function of $\mathbf{r}'$, of course, since no expectation value has been taken yet. However, when we do take an expectation value over momentum, this factor *will* be a function of $\mathbf{r}'$. The kernel measures the strength of the influence of one boid on another. Therefore, we take to calling $K^i{}_j(\mathbf{r})$ the *influence kernel*.

By the convolution theorem,

$$\hat{\eta}^i(\mathbf{r}) = \int \frac{d^d k}{(2\pi)^d} \, e^{i\mathbf{k}\cdot\mathbf{r}} \widetilde{K}^i{}_j(\mathbf{k}) \tilde{\hat{p}}^j(\mathbf{k}), \tag{C.3}$$

where $\widetilde{K}^i{}_j(\mathbf{k})$ and $\tilde{\hat{p}}^j(\mathbf{k})$ are the Fourier transforms of $K^i{}_j(\mathbf{r})$ and $\hat{p}^j(\mathbf{r}')$, respectively. Finally, we may write this as

$$\hat{\eta}^i(\mathbf{r}) = \widetilde{K}^i{}_j(-i\nabla) \int \frac{d^d k}{(2\pi)^d} \, e^{i\mathbf{k}\cdot\mathbf{r}} \tilde{\hat{p}}^j(\mathbf{k}) = \widetilde{K}^i{}_j(-i\nabla) \hat{p}^j(\mathbf{r}). \tag{C.4}$$

Since the hydrodynamic equations are a gradient expansion, we would naturally Taylor-expand $\widetilde{K}^i{}_j$ around the origin. Let us make some important remarks:

1. The expression for $\hat{\eta}^i$ is formal as it only makes sense *after* an expectation value is taken.

2. The influence kernel will naturally come with at least one length scale. For example, in the Vicsek model, each boid tries to align its direction with all the boids within a fixed radius $r_0$ of itself:

$$K^i{}_j(\mathbf{r}) = \frac{\kappa}{\pi r_0^2} \Theta\left(1 - \frac{r}{r_0}\right) \delta^i_j, \tag{C.5}$$

where $\Theta$ is the Heaviside theta function and $\kappa$ is some dimensionless constant setting the normalization of the influence kernel.[26]

Dimensional analysis implies that every gradient factor must be multiplied by a length scale factor. For the kernel given in (C.5), for example, we know that $\widetilde{K}^i{}_j$ is not just a function of $-i\nabla$, but rather of $-ir_0\nabla$, for the same reason that $K^i{}_j$ is not just a function of $\mathbf{r}$, but of $\frac{\mathbf{r}}{r_0}$.

3. We may interpret the influence kernel as the Green's function for some spatial differential operator $\mathcal{O}^i{}_j(\nabla)$:

$$\mathcal{O}^i{}_k(\nabla) K^k{}_j(\mathbf{r}) = -\kappa \delta^i_j \delta(\mathbf{r}), \tag{C.6}$$

where $\kappa$ is a constant with the same dimensions as $\mathcal{O}^i{}_j$. The operator $\mathcal{O}^i{}_j$ and the constant $\kappa$ may be rendered dimensionless by absorbing dimensions using a localization scale $r_0$, as done in (C.5). For instance, the example discussed in sec. 3 is the Green's function for the (dimensionful) Helmholtz operator, $\mathcal{O}^i{}_j(-i\nabla) = \delta^i{}_j(\nabla^2 + \mu^2)$, which gives the famous Yukawa potential. As discussed in sec. 3, $\kappa$ also sets the fluctuation scale that allows this formalism to capture stochastic effects, such as diffusion, without

---

[26]The form of this influence kernel is morally correct even though the force (C.1) is not *quite* the same as that of the Vicsek model. The latter assumes that the boids all have the same speed and that boid "$a$" aligns its orientation $\theta_a$ with boid "$b$" via an interaction of the form $\sin(\theta_b - \theta_a)$. Thus, our starting ansatz for the force is not quite analogous to that of the Vicsek model. Nevertheless, the step-function form of the influence kernel, *is* correct.

the need to introduce noise terms in the force. Indeed, at lowest non-trivial order in spatial gradients, the gradient expansion of $\widetilde{K}^i{}_j(-i\nabla)$ generates the two diffusion terms in the Toner-Tu equation: $\nabla^2 v^i$ and $\nabla^i(\nabla \cdot \mathbf{v})$.

Let us take a moment to appreciate how remarkable this last statement is. Under coarse-graining, the Vicsek model eventually leads to a Toner-Tu-like equation, but with highly constrained coefficients (see [39], specifically Problem 9.2). In particular, it is the *noise* term (a temporally-random force) that ends up producing a diffusion term in a Toner-Tu-like equation. In contrast, using kinetic theory and the simple ansatz of local mean-field momentum matching, we have shown that such diffusion terms already arise due to the choice of influence kernel – no random forces needed.

## C.1 Influence kernels and diffusion coefficients in 2d

We will show how to relate the diffusion coefficients $D_T$ and $D_B$ to the parameter $\lambda$ in the force ansatz (C.1) and a choice of the influence kernel. As a reminder, these diffusion coefficients appear in the Toner-Tu equation as follows:

$$\partial_t v^i + \cdots = \cdots + D_T \nabla^2 v^i + D_B \nabla^i \nabla_j v^j \,. \tag{C.7}$$

As such, they have units of area per unit time, as usual. Therefore, if a single length scale, $r_0$, appears in the influence kernel, then we can already conclude that

$$D_{T,B} \propto \lambda r_0^2 \,. \tag{C.8}$$

Consider, for example, the influence kernel in (C.5). Its Fourier transform is

$$\widetilde{K}^i{}_j(k) = \frac{2\kappa J_1(r_0 k)}{r_0 k} \delta^i_j \,, \tag{C.9}$$

whose Taylor expansion around $k = 0$ is

$$\widetilde{K}^i{}_j(k) = \left(1 - \frac{1}{8}(r_0 k)^2\right)\kappa \delta^i_j + O(r_0 k)^4 \,, \tag{C.10}$$

and thus, dropping all higher gradient terms,

$$\hat{\eta}^i = \left(1 + \frac{r_0^2}{8}\nabla^2\right)\kappa \hat{p}^i \,. \tag{C.11}$$

Finally, the force is

$$\hat{F}^i = \frac{\lambda \kappa r_0^2}{8}\nabla^2 \hat{p}^i \,. \tag{C.12}$$

Converting to velocity gives

$$\hat{F}^i = \frac{\lambda \kappa r_0^2}{8}\left[\frac{\rho}{n}\nabla^2 \hat{v}^i + 2\nabla^j\left(\frac{\rho}{n}\right)\nabla_j \hat{v}^i + 2\hat{v}^i \nabla^2\left(\frac{\rho}{n}\right)\right]. \tag{C.13}$$

Note that $\rho$ should really be $\hat{\rho}$, but as in the rest of this paper, we have assumed that we can pull $\hat{\rho}$ out of any expectation value and simply replace it with its own expectation value, $\rho$. Note that the terms involving spatial gradients of the quantity $\frac{\rho}{n}$ vanish if $\rho = mn$, in the Galilean boost-invariant case. In the Lorentz-invariant case, these constitute third- and higher-order terms in the velocity. In either case, the leading-order result is proportional to the Laplacian of the velocity, which is the $D_T$ term in the Toner-Tu equation. However, the possibility of a more complicated $\frac{\rho}{n}$ factor in the boost non-invariant case certainly allows for the spatial gradients of this factor to appear in the force term.

Table 2: Four choices of isotropic influence kernel in 2d, their Fourier transforms, and the corresponding diffusion coefficient $D_T$ ($\nabla^2 \mathbf{v}$ term).

| Name | $\frac{1}{\kappa}K(r/r_0)$ | $\frac{1}{\kappa}\widetilde{K}(r_0 k)$ | $D_T$ (units of $\lambda \kappa r_0^2$) |
|---|---|---|---|
| Sharp cut-off | $\frac{1}{\pi r_0^2}\Theta(1-\frac{r}{r_0})$ | $\frac{2}{r_0 k}J_1(r_0 k)$ | $\frac{1}{8}$ |
| Screened $\frac{1}{r}$ | $\frac{1}{2\pi r_0 r}e^{-r/r_0}$ | $\left[1+(r_0 k)^2\right]^{-1/2}$ | $\frac{1}{2}$ |
| Yukawa | $\frac{1}{2\pi}K_0(\frac{r}{r_0})$ | $\left[1+(r_0 k)^2\right]^{-1}$ | $1$ |
| Exponential | $\frac{1}{2\pi r_0^2}e^{-r/r_0}$ | $\left[1+(r_0 k)^2\right]^{-3/2}$ | $\frac{3}{2}$ |
| Gaussian | $\frac{1}{2\pi r_0^2}e^{-\frac{1}{2}(r/r_0)^2}$ | $e^{-\frac{1}{2}(r_0 k)^2}$ | $\frac{1}{2}$ |

Focusing just on the diffusion term, we conclude that the $\frac{nF^i}{\rho}$ term in the Euler equation in (B.12) reads

$$\frac{nF^i}{\rho} = \frac{\lambda \kappa r_0^2}{8}\nabla^2 v^i + \cdots , \tag{C.14}$$

and thus, we identify the diffusion coefficients

$$D_T = \tfrac{1}{8}\lambda \kappa r_0^2 , \qquad D_B = 0 . \tag{C.15}$$

The class of influence kernel that can be written as

$$K^i{}_j(\mathbf{r}) = K(r/r_0)\,\delta^i_j , \tag{C.16}$$

will always produce $D_B = 0$. We offer some 2d examples and their corresponding Fourier transforms and $D_T$ values in table 2. Of course, to the order that we are keeping, all sufficiently localized functions are approximately Gaussian with some width.

We can easily generate a $D_B$ term as well if we set the Fourier transform of the influence kernel to be, for example,

$$\widetilde{K}^i{}_j(\mathbf{k}) = \kappa e^{-\frac{1}{2}(r_0 k)^2 \delta^i_j - \chi r_0^2 k^i k_j + O(k^4)} , \tag{C.17}$$

where $\chi$ is just a real parameter. The Taylor expansion of this would then be

$$\widetilde{K}^i{}_j(\mathbf{k}) \approx \left(1 - \frac{1}{2}(r_0 k)^2\right)\kappa \delta^i_j - \kappa \chi r_0^2 k^i k_j + O(k^4) , \tag{C.18}$$

and so

$$\widetilde{K}^i{}_j(-i\nabla) = \left(1 + \frac{r_0^2}{2}\nabla^2\right)\kappa \delta^i_j + \kappa \chi r_0^2 \nabla^i \nabla_j + O(\nabla^4) , \tag{C.19}$$

which, when acting on $\hat{p}^j$ produces the $D_T$ and $D_B$ terms with

$$D_T = \frac{\lambda \kappa r_0^2}{2} , \qquad D_B = \lambda \kappa \chi r_0^2 . \tag{C.20}$$

The Fourier transform of the influence kernel has a problem, however, since $k^i k_j$ can become arbitrarily negative, which makes the off-diagonal components of $\widetilde{K}^i{}_j$ unbounded. It also vanishes along the axes. There are many ways to remedy this. We will discuss two methods below.

**Method 1:** Add an $O(k^4)$ term, such as $-\epsilon (r_0^2 k^2)^2$, to the off-diagonal components of $\widetilde{K}^i{}_j$. This breaks the nice tensorial structure by a small amount if $\epsilon \ll 1$ in exchange for bounding the off-diagonal components of $\widetilde{K}^i{}_j$. This has the unfortunate consequence that we cannot actually write down the off-diagonal components of the influence kernel in space: without the $\epsilon$ term, the inverse Fourier transform is not well-defined, and with the $\epsilon$ term, the inverse Fourier transform does not have a closed analytic form. Nevertheless, we can write down the diagonal components of the influence kernel in space:

$$K^x{}_x(\mathbf{r}) = \kappa \, \frac{e^{-\frac{1}{2(1+2\chi)}(\frac{x}{r_0})^2}}{\sqrt{2\pi(1+2\chi)}\, r_0} \cdot \frac{e^{-\frac{1}{2}(\frac{y}{r_0})^2}}{\sqrt{2\pi}\, r_0}\,, \tag{C.21a}$$

$$K^y{}_y(\mathbf{r}) = \kappa \, \frac{e^{-\frac{1}{2}(\frac{x}{r_0})^2}}{\sqrt{2\pi}\, r_0} \cdot \frac{e^{-\frac{1}{2(1+2\chi)}(\frac{y}{r_0})^2}}{\sqrt{2\pi(1+2\chi)}\, r_0}\,. \tag{C.21b}$$

These are simply elliptical Gaussian influence kernels that are wider in the $x$-direction when matching the $x$-component of momentum and wider in the $y$-direction when matching the $y$-component of the momentum. This directional anisotropy is controlled by the parameter $\chi$ and is necessary to produce a $D_B$ term: both anisotropy and $D_B$ vanish in the $\chi \to 0$ limit.

**Method 2:** A simpler method would be to write $\widetilde{K}^i{}_j$ as

$$\widetilde{K}^i{}_j(\mathbf{k}) = \kappa\left(\delta^i_j - \chi r_0^2 k^i k_j\right) e^{-\frac{1}{2}(r_0 k)^2}\,, \tag{C.22}$$

instead of (C.17). In real space, this is a superposition of an isotropic $\delta^i_j$ term as well as an anisotropic Hessian-type term:[27]

$$K^i{}_j(\mathbf{r}) = \frac{\kappa}{2\pi r_0^2}\left[\delta^i_j + \chi r_0^2 \frac{\partial}{\partial r_i} \frac{\partial}{\partial r^j}\right] e^{-\frac{1}{2}(\frac{r}{r_0})^2}\,, \tag{C.23}$$

which evaluates to

$$K^i{}_j(\mathbf{r}) = \frac{\kappa}{2\pi r_0^2}\left[(1-\chi)\delta^i_j + \chi \frac{r^i r_j}{r_0^2}\right] e^{-\frac{1}{2}(\frac{r}{r_0})^2}\,. \tag{C.24}$$

If the influence kernel had been normalized ($\kappa = 1$), then the would-be leading term in $\hat{\mathbf{F}}$, which is linear in momentum, would vanish leaving us with the diffusion terms at lowest order in the gradient expansion. With $\kappa \neq 1$, there is a residual force term linear in momentum. Thus, we might as well just fold this into a general *linear drag force* and redefine our force ansatz to be

$$\hat{\mathbf{F}} = -\frac{\hat{\mathbf{p}}}{\tau'} - \lambda(\kappa \hat{\mathbf{p}} - \hat{\boldsymbol{\eta}})\,, \tag{C.25}$$

where $\tau'$ is a characteristic relaxation time scale associated with a linear drag force. In the next section, we consider the remaining effects of this force on the Euler equation.

---

[27]Thanks to the referee for pointing out the possibility of a Hessian term in the influence kernel.

## C.2 Linear drag and the remaining force terms

To compute the contributions of the remaining force terms in (B.12), we have to go back to equation (B.10) and compute the last two force terms:

$$2\tau n\,\overline{\hat{v}^{(i}\hat{F}^{j)}} = -\frac{2\tau}{\tau'}n\,\overline{\hat{v}^{(i}\hat{p}^{j)}} = -\frac{2\tau}{\tau'}\rho\,\overline{\hat{v}^i\hat{v}^j}\,, \tag{C.26a}$$

$$\tau\rho\,\overline{\hat{v}^i\hat{v}^j\big(\nabla_{\hat{\mathbf{p}}}\cdot\hat{\mathbf{F}}\big)} = -\frac{d\tau}{\tau'}\rho\,\overline{\hat{v}^i\hat{v}^j}\,. \tag{C.26b}$$

Remember that we eventually have to take a spatial derivative of these terms (contracted with the $j$ index). Therefore, plugging in either of the diffusion terms in the force into the above two terms will give higher-order terms in the hydrodynamic expansion (quadratic in velocity and cubic in spatial gradients). Hence, we ignore those terms.

Plugging (C.26) into (B.10) gives

$$\rho\,\overline{\hat{v}^i\hat{v}^j} = \frac{\rho\big(\overline{\hat{v}^i\hat{v}^j}\big)_{\text{eq}}}{1+(d+2)\frac{\tau}{\tau'}}\,. \tag{C.27}$$

It now makes sense to posit the equilibrium relation

$$\big(\overline{\hat{v}^i\hat{v}^j}\big)_{\text{eq}} = \left(1+(d+2)\frac{\tau}{\tau'}\right)\left(v^iv^j + \frac{P}{\rho}\delta^{ij}\right), \tag{C.28}$$

which is just (30) multiplied by the factor which depends on the two time constants $\tau$ and $\tau'$. The end result is to effectively remove these two force terms altogether as their only effect is to slightly increase the standard deviation of velocity. Note that even if we had not absorbed the factor of $1+(d+2)\frac{\tau}{\tau'}$ into the equilibrium velocity two-point function, that factor is very close to unity anyway since $\tau \ll \tau'$. That is, the collision time in the phase space of *states* of the system should be much less that the collision time for boid-environment collisions that lead to the drag force.

Only one other force term remains:

$$\overline{\hat{v}^i\big(\nabla_{\hat{\mathbf{p}}}\cdot\hat{\mathbf{F}}\big)} = -\frac{d}{\tau'}v^i\,. \tag{C.29}$$

These lead to the result

$$\begin{aligned}\partial_t v^i + \big(\mathbf{v}\cdot\nabla\big)v^i &= -\frac{\partial^i P}{\rho} - \frac{v^i}{\rho}\big[\partial_t\rho + \nabla\cdot(\rho\mathbf{v})\big] - \left(\frac{1}{\tau}+\frac{d+1}{\tau'}\right)v^i\\ &\quad + \frac{v^i_{\text{eq}}}{\tau} + D_T\nabla^2 v^i + D_B\nabla^i\big(\nabla\cdot\mathbf{v}\big).\end{aligned} \tag{C.30}$$

Again, introducing the equation of kinetic state as in (44), we are left precisely with the Toner-Tu equation

$$\begin{aligned}\partial_t\mathbf{v} + \lambda_1(\mathbf{v}\cdot\nabla)\mathbf{v} + \lambda_2(\nabla\cdot\mathbf{v})\mathbf{v} + \lambda_3\nabla(v^2) &= (\alpha-\beta v^2)\mathbf{v} - \frac{\nabla P_0}{\rho_0} - \frac{\mathbf{v}-\mathbf{v}_{\text{eq}}}{\tau}\\ &\quad + D_T\nabla^2\mathbf{v} + D_B\nabla(\nabla\cdot\mathbf{v}),\end{aligned} \tag{C.31}$$

with the various parameters given in (51) and (52), except that the $\alpha$ identification is modified slightly to

$$\alpha = -g\big|_{\nabla\cdot\mathbf{v}=v^2=0} - \frac{d+1}{\tau'}\,, \tag{C.32}$$

and where the diffusion coefficients are given by the parameter $\lambda$ and the parameters of the influence kernel, as in (C.20), for example.

# D   On derivative expansions and stochastic fluctuations

In this appendix we illustrate how derivative expansions encode short distance stochastic fluctuations at long wavelengths in any statistical or field theoretic formalism. Although implicit in the rationale of particle physics effective field theory, where a derivative expansion constitutes a functional Taylor expansion that encapsulates the short distance physics one has effectively coarse-grained or integrated out [68], an explicitly worked out example can be informative to the point of novelty. The discussion that follows draws from a problem in the electrostatic limit of quantum electrodynamics [69, 70], and although it features quantum fluctuations, the formalism directly generalizes to any statistical field theory where the moment generating functional can be associated with a partition function for a continuum of degrees of freedom.

Consider scattering a point charge off of another, infinitely heavy point charge each with charge $e_0$, so that the relativistic momentum transfer is given by $k^\mu = (0, \mathbf{k})$. The electrostatic potential is given by the inverse Fourier transform of the momentum space scattering amplitude [69]:

$$V(r) = \frac{e_0^2}{4\pi} \int \frac{d^3 k}{(2\pi)^3} e^{i\mathbf{k}\cdot\mathbf{r}} D_F'(k) \bigg|_{k_0=0} = \hbar c \int \frac{d^3 k}{(2\pi)^3} e^{i\mathbf{k}\cdot\mathbf{r}} \frac{\alpha(\mathbf{k}^2)}{\mathbf{k}^2}, \tag{D.1}$$

where $D_F'(k)$ is defined via the full photon Feynman propagator $D_F'^{\mu\nu}(k) = g^{\mu\nu} D_F'(k)$, given by

$$i D_F'^{\mu\nu}(k) := i D_F^{\mu\nu}(k) + i D_F^{\mu\lambda}(k) i\Pi_{\lambda\kappa}(k) i D_F^{\kappa\nu}(k) + \cdots, \tag{D.2}$$

where the insertions $\Pi_{\lambda\kappa}(k)$ convey the effects of loops of virtual electron-positron pairs on photon propagation, and whose resummation realizes the effects of vacuum polarization. In terms of a heuristic physical picture: Virtual electron-positron pairs are constantly 'appearing and disappearing' from the vacuum,[28] effectively manifesting as a stochastic background of fluctuating dipoles. The latter is responsible for screening effects as well as Casimir forces in bounded geometries, and whose effects on photon propagation correspond to the insertions in (D.2). The free Feynman propagator that appears in (D.2) is given in Lorentz gauge by:

$$D_F^{\mu\nu}(k) = -\frac{\eta^{\mu\nu}}{k^2 + i\varepsilon}, \tag{D.3}$$

where $\eta^{\mu\nu}$ is the inverse Minkowski space metric tensor. Note that in position space, the time ordered correlation function of the photon field can be expressed as

$$G_F^{\mu\nu}(x, x') := \langle \mathrm{T}\{A^\mu(x) A^\nu(x')\} \rangle_0 = i\hbar c D_F^{\mu\nu}(x, x'), \tag{D.4}$$

where the subscript on the angled brackets denotes vacuum expectation value. Given the Fourier representation (D.3), one finds that

$$\Box_x G_F^{\mu\nu}(x, x') = -i\hbar c \, \delta^4(x, x') \eta^{\mu\nu}, \tag{D.5}$$

where $\Box_x$ is the d'Alembertian operator with respect to the $x$ coordinate (to be compared with (37) and (40) with $-i G_F^{ij}$ identified with $K^{ij}$ and with $\kappa$ identified as $\hbar c$, notwithstanding the different dimensions in which these expressions are defined).

If the fine structure constant were taken to be a constant, such that $4\pi\alpha \equiv e_0^2/(\hbar c)$ where $e_0$ is the electron charge measured at large charge separation, then one can immediately perform the Fourier integral to obtain

$$V(r) = \frac{e_0^2}{4\pi r}, \tag{D.6}$$

---

[28]Which can be thought of as the reservoir from which quantum fluctuations (in this case, virtual pairs) borrow energy, only to have to repay it within the window demanded by the energy-time uncertainty relation, and is akin to canonical ensemble thermal fluctuations of a system coupled to a reservoir with $\hbar$ as the analog of $\beta^{-1}$.

which corresponds to the potential between two point charges, each with charge $e_0$. However, we know that the corrections to the photon propagator (D.2) amount to the running coupling [71]:

$$\alpha \to \alpha\left(\mathbf{k}^2\right) = \alpha_0 \left[1 + \frac{\alpha_0}{3\pi} \log \frac{\hbar^2 \mathbf{k}^2}{m_e^2 c^2} + \mathcal{O}\left(\alpha_0^4\right)\right], \qquad \hbar^2 \mathbf{k}^2/(m_e^2 c^2) \gg 1, \tag{D.7}$$

where $4\pi\alpha_0 \equiv e_0^2/(\hbar c)$. In the context of (D.1), this can be viewed in one of two operationally equivalent ways. Either, one can choose to preserve the Gauss law, which relates the potential to the charge density as $\nabla^2 V = -\rho$, so that what was a point charge now gets smeared via quantum effects to the charge distribution

$$\rho_q(\mathbf{r}) = \int \frac{d^3 k}{(2\pi)^3} e^{i\mathbf{k}\cdot\mathbf{r}} \alpha\left(\mathbf{k}^2\right), \tag{D.8}$$

or, one can instead rewrite (D.1) as

$$V(r) = \hbar c \int \frac{d^3 k}{(2\pi)^3} \frac{\alpha_0}{\mathbf{k}^2} \left[1 + \frac{\alpha_0}{3\pi} \log\left(\frac{-\hbar^2\nabla^2}{m_e^2 c^2}\right) + \dots\right] e^{i\mathbf{k}\cdot\mathbf{r}}, \tag{D.9}$$

and keep the charge distribution as point source but instead, consider the Gauss law to get derivative corrections of the form:

$$\left[1 - \frac{\alpha_0}{3\pi} \log\left(\frac{-\hbar^2\nabla^2}{m_e^2 c^2}\right) + \dots\right] \nabla^2 V = -\rho_c, \tag{D.10}$$

where $\rho_c$ denotes the classical distribution and $\rho_q$ denotes the quantum mechanically smeared distribution. Both are entirely equivalent, and are constructed to result in the quantum corrected electrostatic potential [69, 71]:

$$V(r) = \frac{e_0^2}{4\pi r} \left[1 + \frac{\alpha_0}{3\pi} \left(\ln \frac{\hbar^2}{m_e^2 c^2 r^2} - 2\gamma_E - \frac{5}{3}\right) + \dots\right], \tag{D.11}$$

where $\gamma_E$ is the Euler-Mascheroni constant, and whose gradients are physically observable field strengths.

Both (D.8) and (D.10) illustrate how the substance of quantum mechanics – vacuum polarization effects and the smearing of sources – is conveyed though a derivative expansion which captures the long wavelength effects of microscopic stochastic fluctuations in an interacting theory. The same will be true in a statistical field theory, where the 'free' theory corresponds to some ensemble where all higher order correlations vanish, and the interactions are given by the moment expansion of the generating functional that generates these correlations.

# E   Physical interpretation of the equation of kinetic state

In this appendix, we expand upon the concept of *inertial mass*, defined as resistance to acceleration under the application of external forces, how it gives rise to the *kinetic mass density*, as well as the notion of an *equation of kinetic state* as an additional constitutive relation in any system where boost symmetries are absent.

If a particle or boid exchanges momentum with its environment and we wish to take account of this exchange without having to model the specific mechanisms underlying it, then we can simply fold in the a priori unknown forces involved in that underlying mechanism into an effective inertial mass. Heuristically, one can do this by setting the net *external* force equal to $m_{\text{in}}\mathbf{a}$, where the subscript abbreviates *inertial*. Since the forces involved in the exchange of

momentum between the boid and the environment may be functions of velocity, the inertial mass will also be a function of velocity in general. In the language of effective field theory, we have essentially *integrated out* the degrees of freedom of the environment leading to a modification of the properties of the boid itself, in this case, its effective mass. Any operator expansion of the interactions with the environment will result in a series of contributions to the effective mass for any given propagating mode.

This is, of course, intimately familiar to the condensed matter physicist who does not work in a non-interacting vacuum, but rather, in a complicated material in which the effective mass of a particle must always take into account its interactions with the environment (e.g., heavy fermions or even heavy photons à la the Meissner effect). In the simple case in which the net exchange force is independent of and simply added to the external force, all we have to do is factor $F_{\text{ext}}$ out of the "force" side of Newton's second law and divide out by the factor containing the exchange force, denoted $F_{\text{exc}}$, which would give the inertial mass

$$m_{\text{in}} = \frac{m}{1 + \frac{F_{\text{exc}}}{F_{\text{ext}}}} \,. \tag{E.1}$$

Of course, propagation in media can be much more complicated in general, so we reconsider this by first working in the simplified setting where the exchange forces are in principle specified and easy to model, in which case one can write down an explicit form for the inertial mass.

Consider first the example of a mass $m$ moving in one dimension in a fluid under the influence of a constant external force $F$, and a drag force due to its interaction with the fluid itself. In this case, the drag force is the force involved in the exchange of momentum between the mass and its fluid environment. The simplest model for this exchange is one in which we suppose the fluid molecules are initially at rest and the mass collides elastically with those fluid molecules as it moves, thereby transferring momentum to its fluid environment and losing momentum in the process. The number of fluid molecules with which the mass collides in a short period of time $dt$ is proportional to its speed $v$ and the amount of momentum that it transfers to a fluid molecule at each collision is proportional to $v$ as well. Therefore, this simple model predicts that the drag force scales quadratically with the speed of the moving mass. At low Reynolds number (e.g., at low speed or high viscosity), semi-phenomenological arguments will conclude that the coefficient between the drag force and the $v^2$ factor arising by the aforementioned exchange mechanism can itself scale *inversely* with speed, and therefore give a net *linear* dependence of the drag force on velocity.[29] This leads to the standard rule of thumb that drag force tends to scale linearly, $\mathbf{F}_{\text{exc}} \equiv \mathbf{F}_{\text{drag}} = -\alpha \mathbf{v}$, at low velocity and quadratically, $\mathbf{F}_{\text{drag}} = -\beta v \mathbf{v}$, at high velocity. The inertial mass is

$$m_{\text{in}} = \frac{m}{1 - \frac{\alpha v}{F}} \,, \qquad \text{or} \qquad m_{\text{in}} = \frac{m}{1 - \frac{\beta v^2}{F}} \,. \tag{E.2}$$

As expected, the inertial mass increases as the velocity increases and evidently diverges as the velocity approaches the terminal velocity, at which point the acceleration vanishes. This is of course a rather naïve picture, and in actual fact, the velocity dependence becomes more complicated and can surpass the terminal velocity nominally inferred from interactions at low momentum exchange as one approaches on the onset of shock formation. Nevertheless, one can infer a great deal by proceeding phenomenologically.

---

[29]In the simplest case where the environment can be modeled as an ensemble of decoupled harmonic oscillators (i.e. is itself close to equilibrium) which interacts with system constituents via linear couplings, one can explicitly calculate a linear drag force [72, 73]. More complicated interactions organized in terms of power counting relevance will generate more complicated dependence on the velocity and its derivatives.

Modeling the momentum exchange of a particle or boid with its environment simplifies greatly if one assumes that any effective drag force from momentum exchange is collinear to the applied force (i.e. one can ignore 'magnetic' or curl type effective interactions). Hence, presuming the applied force to be along any arbitrary fixed direction $\mathbf{F} \equiv F\,\hat{\mathbf{i}}$, so that $\mathbf{F}_{\text{drag}}$ can also be expressed as $\mathbf{F}_{\text{drag}} = -\zeta(v)\,\hat{\mathbf{i}}$, where $v$ is defined via $\mathbf{v} = v\,\hat{\mathbf{i}}$, one obtains the applied force equation and corresponding effective inertial mass for a particle of mass $m$ to be:

$$m\dot{v} = F - \zeta(v)\,, \qquad m_{\text{in}} = \frac{m}{1 - \frac{\zeta(v)}{F}}\,, \tag{E.3}$$

where the above will result independent of the direction of the applied force. In this specific example, where the constituents of the fluid all experience the the same external and drag forces, the kinetic mass density can be identified as the product of the particle number density, $n$, and the inertial mass of each particle,

$$\rho = m_{\text{in}} n\,. \tag{E.4}$$

Of course, in the most general case, $\rho$ is treated more abstractly as a thermodynamic variable whose relationship with the microscopic quantities such as particle number density can be far more complicated than in this simple example. The velocity dependence of $\rho$ immediately follows as the most general possibility for any system where momentum can be exchanged with the reservoir.

It is straightforward to derive the non-conservation equations for $\rho$ via the conservation of number density (18) and the defining relation for the equation of kinetic state:

$$\partial_t \rho + \nabla \cdot (\rho \mathbf{v}) = g\rho\,, \tag{E.5}$$

through which one obtains

$$g \equiv D_t \log m_{\text{in}}\,, \tag{E.6}$$

where $D_t$ is defined as the convective derivative $\partial_t + \mathbf{v} \cdot \nabla$, and where we have used both expressions in (E.3). When one can neglect gradients, one obtains the simple expression

$$g = \frac{\zeta'(v)}{m}\,. \tag{E.7}$$

Note that $g$ can be a function of $v$ in this case (not just $v^2$), and where the absence of any term that would correspond to $\nabla \cdot \mathbf{v}$ corresponds to being able to neglect the gradient term in the convective derivative in the defining expression (E.6). More complicated functional forms can of course arise were one to have derivative couplings entering the operator expansion of the environmental interactions, in addition to local variations. In any event, we see the starting point of a formal argument for the vanishing of $\lambda_2$ in (52) – such terms correspond to sub-leading interactions in the power counting sense, and should become increasingly irrelevant at long wavelengths.

In closing our discussion, we note that we have presumed that boids are neither born nor die mid-flow and so the boid particle number density is conserved, even if the kinetic mass density in general is not. One may look to revist the assumptions placed on our collision terms as we derived the Vlasov hierarchy, especially in cases where some form of agency is involved in the exchange of momentum with the environment, as with more interesting examples of active systems. How and when a boid chooses to perform the exchange depends crucially on the current state of the system, though perhaps only locally in a neighborhood of the boid. It is likely very difficult, if not altogether impossible, to model this exchange mechanism exactly taking agency into account. Of course, one hopes that agency, insofar as it may involve es-oteric issues such as free will or even consciousness, is essentially irrelevant and serves only

one purpose: to enact *whatever* exchange mechanism is necessary to achieve a goal, such as maintaining formation in a flock. For example, the Vicsek model does not care about how a boid steers itself in order to align with the other boids in its neighborhood, only that it does. In such cases, one wishes to avoid having to model the microscopic interactions involved in the exchange mechanism itself. What we have demonstrated in this paper is that all of these interaction details can be neatly packaged in three quantities: the kinetic mass density $\rho$, the function $g$ in the kinetic equation of state which parametrizes the non-conservation of $\rho$, and the influence kernel. From these ingredients, one can derive the hydrodynamics of an interesting subset of active flocks purely from a kinetic theory analysis.

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
