# Peer review of "Hydrodynamics without Boost-Invariance from Kinetic Theory: From Perfect Fluids to Active Flocks"

_SciPost Physics, doi:SciPost Phys. 19, 071 (2025)_

## Round 2 · Referee Report · Anonymous (Referee 2) · 2025-4-15

Strengths

This paper addresses research directions that are both fundamental and applied.

1- On the conceptual side, it highlights the fact that most systems break boost symmetries, making it necessary to develop non-boost-invariant theories—precisely the aim of this work.

2- The paper also establishes connections with well-established theories of flocking hydrodynamics, refining our understanding of these models.

3- The mathematical developments are clearly presented, with many technical details provided in the appendices. Both the results and the outlook are insightful and thoroughly discussed.

Weaknesses

Given its topic, this paper is intended for a diverse range of physics communities. However, the specific terminology associated with flocking theory may occasionally be confusing to readers unfamiliar with that particular subfield.

Report

In this paper, the authors derive the kinetic theory of a spatially isotropic classical system without assuming either Galilean or Lorentz boost invariance. Their starting point for the statistical analysis is the Boltzmann equation, where the collision functional is generic and satisfies certain axioms (consistent with the boost-invariant cases). This collision functional accounts for both inter-particle collisions and interactions between the fluid and the environment. The latter is responsible for momentum exchange. Other inter-particle forces, referred to here as boid forces, are incorporated on the left-hand side of Boltzmann’s equation. The statistical analysis is carried out using the Vlasov hierarchy up to third order in velocity moments. From this, the hydrodynamic equations are derived, with the main signature of non-boost invariance being that the canonical momentum is no longer equal to the kinetic momentum (or its relativistic Lorentzian counterpart).

From the hydrodynamic equations, the authors:

1 - Successfully recover the non-boost-invariant hydrodynamics of an ideal fluid by setting the boid forces to zero and imposing detailed balance (i.e., a vanishing collision functional).

2 - Discuss the case where boid forces tend to align particle velocities with a local average velocity field, inspired by the Vicsek model.

3 - Recover the Toner-Tu theory by adopting the Bhatnagar-Gross-Krook-Welander (BGKW) form for the collision functional, setting the boid forces to zero, and introducing the notion of an equation of kinetic state to effectively capture the relation between canonical momentum and velocity. This top-down approach allows the authors to extract information about the parameters of the Toner-Tu theory and to provide symmetry-based explanations for the technical naturalness of some of these parameters.

Requested changes

This work is detailed, compelling, and insightful. Before recommending it for publication, I would like to raise a few questions, comments, and suggestions:

1 - Below equation (11), it is stated that the interactions with the environment are included in the collision functional of Boltzmann’s equation. It is then mentioned that the BGKW expression will be used for the collision functional. To the best of my knowledge, the BGKW expression is based on the physical interpretation that the collision term describes the rate at which collisions change the distribution function over time. The explicit BGKW form assumes that the duration of collisions is much shorter than the relaxation time. For inter-particle interactions, this corresponds to requiring that the interaction range is much smaller than the typical inter-particle distance (i.e., a dilute gas of “little balls”). What, then, are the requirements on the interactions with the environment for the BGKW approximation to remain valid?

2 - Related to question 1: below equation (13), $\tau$ is referred to as a “collision time.” Shouldn’t it instead be interpreted as a relaxation time—the typical time between “instantaneous” collisions?

3 - In equation (30), the most general expression is given without gradients. Is the reason for excluding gradients that the left-hand side corresponds to an equilibrium (ideal fluid) situation?

4 - Below equation (36), the phrase “when supplemented with spatial translation invariance of the system” is used. However, due to the presence of interactions with the environment, the system exchanges momentum with its surroundings. I would therefore naively expect that spatial translation is not a symmetry of the system, since momentum is not conserved. Should this statement instead refer specifically to the spatial translational invariance of the boid forces?

5 - Throughout the paper, isotropy is used to argue that certain quantities—particularly the kinetic mass density—scale as $v^2$. Why couldn’t they instead depend on the norm of the velocity?

6 - (Optional) For clarity, it would be helpful to either add a reference or provide a brief physical explanation for why truncating the Vlasov hierarchy at third order in velocity momenta is a good approximation.

7 - (Optional) To improve accessibility, it may be beneficial to define the specific vocabulary of flocking theory in the Introduction, in order to prevent non-specialist readers from misinterpreting key concepts. For example, in the Introduction, just below equation (4), the phrase “where inter-boid forces are presumed to vanish” could be misleading for non-specialists, as inter-boid forces might be naively interpreted as collisions between particles. This could cause confusion, since equation (4) still contains a (BGKW) collision term. That said, the meaning becomes clear once the full set of computations is presented.

Recommendation

Ask for minor revision

---

## Round 2 · Referee Report · Anonymous (Referee 3) · 2025-6-18

Strengths

  1. Original results

  2. Clearly written.

  3. Of interest to several communities.

Weaknesses

  1. Comparison with existing literature not thorough enough

  2. Some of the main results may not be especially novel.

Report

This paper presents a kinetic theory derivation and generalization of the boost non-invariant hydrodynamic equations introduced in ref [1]. Through a derivative expansion of interactions the equations match with the Toner Tu equations used to describe active matter, with values of the coefficients determined by symmetries and thermodynamic relations.

The main ingredient is the introduction of a kinetic mass density that determines a relation between velocity and momentum. There are some assumptions concerning the expectation values of product of operators that are introduced to simplify the kinetic theory derivation, but that seem reasonable and on par with related literature.

Overall the paper looks very solid technically and presents new results, so in my opinion it should be published. There are just some of questions/comments that I think should be addressed by the authors before.

Requested changes

1) In the original Toner and Tu paper (ref [20]), the hydrodynamic equations (1) in their paper take the form in equation (50) with the values for the coefficients (53). There is a foonote ([6]) stating that $\lambda_1\neq 1$ and $\lambda_2\neq 0$ are irrelevant in the IR. This seems to be a very similar statement to the one the authors are making in the paper. In the Toner and Tu paper the demonstration is deferred to another publication. The authors probably should mention this and check if such derivaiton exists, and cite it accordingly if it is the case.

2) This referencehttps://arxiv.org/abs/1006.1825v2 derives Toner Tu equations from Vicsek model using kinetic theory. It is probably worth comparing its results with the results of this paper and checking if there are other similar works.

3) In equation (37) rotational invariance would also allow for a contribution to the kernel of the form $K^i_j=\partial^i \partial_j \phi$, with $\phi$ a scalar function. Depending on what $\phi$ is, this is not necessarily suppressed in the derivative expansion. How would this affect to the discussion? A related question is whether parity invariance is assumed in the whole derivation, or if it could be broken without introducing new terms.

Recommendation

Publish (easily meets expectations and criteria for this Journal; among top 50%)

---

## Round 3 · Author Response

Thank you very much for your taking the time to read our manuscript carefully and for your valuable comments and suggestions. We have endeavored to incorporate your suggestions and reply to your questions, making related changes to the text when appropriate. These changes are catalogued below. We submit our revised manuscript for your review.

---

## Round 3 · List of Changes

1. Page 3: Promoted what used to be footnote 2 (origin of the term boid) to the main text.

  2. New footnote 7 states that "collision time" approximation may also be called "relaxation time" approximation, but that this does not just mean low number density, but also depends on the interaction strength.

  3. Augmented what used to be footnote 9 to clarify the enslavement procedure to terminate the Vlasov hierarchy.

  4. Page 9: Comment on the standard expression for the equilibrium expectation value of the product of two velocities. This holds in the bulk, away from boundaries.

  5. Page 10: Added clarifying statements about the influence kernel - it captures only the inter-boid interactions, which are assumed to be translation-invariant and isotropic.

  6. Added footnote 15 stating the assumption of the analyticity of the small-v expansion.

  7. Page 14: Added a statement that the renormalization group analysis of Toner-Tu theory is not fully resolved and cite the four recent relevant works.

  8. Page 14: Added a statement that noise can be added by hand, as is done in Toner-Tu theory.

  9. Page 16: In response to Referee #3, added a discussion about work by Thomas Ihle deriving Toner-Tu theory from a coarse-graining of the Vicsek model.

  10. Page 23: Added a Method 2 for how to solve the problem of modifying the influence kernel to generate diffusion coefficients while remaining well-defined in real space. This was inspired by the question from Referee #3 about Hessian terms in the influence kernel and so we thank the referee in footnote 27.

  11. Page 28: Corrected two typos: 1) the numerator of m_{in} in (E3) should be m, not 1; and 2) the term (v.nabla)rho in (E5) should be nabla.(rho v).

---

## Editorial Decision

published